# ExGRG: Explicitly-Generated Relation Graph for Self-Supervised Representation Learning

## Abstract

Self-supervised Learning (SSL) has emerged as a powerful technique in pre-training deep learning models without relying on expensive annotated labels, instead leveraging embedded signals in unlabeled data. While SSL has shown remarkable success in computer vision tasks through intuitive data augmentation, its application to graph-structured data poses challenges due to the semantic-altering and counter-intuitive nature of graph augmentations. Addressing this limitation, this paper introduces a novel non-contrastive SSL approach to **Ex**plicitly **G**enerate a compositional **R**elation **G**raph (ExGRG) instead of relying solely on the conventional augmentation-based implicit relation graph. ExGRG offers a framework for incorporating prior domain knowledge and online extracted information into the SSL invariance objective, drawing inspiration from the Laplacian Eigenmap and Expectation-Maximization (EM). Employing an EM perspective on SSL, our E-step involves relation graph generation to identify candidates to guide the SSL invariance objective, and M-step updates the model parameters by integrating the derived relational information. Extensive experimentation on diverse node classification datasets demonstrates the superiority of our method over state-of-the-art techniques, affirming ExGRG as an effective adoption of SSL for graph representation learning.

## 1 Introduction

In supervised deep learning, models are traditionally trained using annotated data. Recognizing the high cost involved, SSL methods capitalize on the abundance of readily available unlabeled data. The underlying principle is to leverage existing signals within the data distribution for pre-training parametric deep models. SSL techniques typically involve applying data augmentations to input samples. An invariance objective is formulated to encourage identical representations for a pair of random views originating from the same source sample.

In computer vision, data augmentations are straightforward and intuitive. They include operations like cropping, rotation, noise adding, and color jittering. On the contrary, the data augmentation in the graph domain is not as intuitive as in vision. Particularly in node classification, common augmentations include node feature masking and edge dropout, which heavily rely on the specific characteristics of the data distribution. Forming representations based on masking certain features may seem counter-intuitive to domain experts, leading to semantic-altering augmentations (Lee et al., 2022). Adding or removing random edges to the source graph can significantly impact the structural properties of the graph (Sun et al., 2021; Lee et al., 2022). Thus, relying on invariance loss solely based on graph augmentation is misleading and lacks the necessary information to learn robust representations.

To alleviate the shortcomings of data augmentation, our work proposes integrating additional appropriate information derived from the input graph. This aids in determining which data samples are candidates to share similar representations. Rather than the prevailing rigid binary determination of whether two points should share a similar representation or not, we embrace a soft evaluation of the degree of similarity. Given the inadequacy and misguidance of graph augmentations, our primary contribution is to integrate additional cues to more effectively guide the determination of analogous representations and assess the relative signifi-

cance of each pairwise comparison. To achieve this, we introduce a novel approach that explicitly generates a higher-order compositional relation graph that pinpoints crucial relation pairs, serving as candidates for the invariance loss. We propose a comprehensive strategy for constructing this compositional relation graph, leveraging insights from three key sources, each contributing to its distinct relation graph: (i) *Neighborhood Similarity in the Representation Space*, (ii) *Higher-Order Graph Encodings*, encompassing the source graph adjacency alongside diverse positional and structural encodings (PSEs), and (iii) *A Deep Clustering* module. We enrich the learning process, elevating the acquired representation and mitigating the deficiencies associated with relying exclusively on augmentations. While we motivate and emphasize the efficacy of our model based on VICReg (Bardes et al., 2021), our approach can be seamlessly integrated with other SSL methods simply by modifying their invariance terms through the exact process.

Graph Neural Networks (GNNs) have the inherent ability to capture label-related patterns due to the message-passing paradigm and the intrinsic characteristics of graph datasets. Hence, we construct a k-Nearest Neighbour (kNN) graph on the representations. Nodes in close proximity within the kNN graph exhibit a heightened likelihood of sharing the same label class. Therefore, this kNN graph serves as our first source to construct the compositional relation graph. PSEs, alongside the adjacency information, constitute our second primary source. Inspired by Lee et al. (2022), we employ the original input adjacency to construct a relation graph. Particularly in homophilic source graphs, adjacent nodes commonly share the same label class. Employing appropriate PSEs, transformers yield results comparable to message-passing networks. When a pair of points shares similar positional and structural properties, it indicates their overall similarity, offering auxiliary guidance for forming meaningful representations. Moreover, we incorporate a deep clustering algorithm rooted in the optimal transport (Asano et al., 2019) as the third source for guiding the invariance term. This module partitions the points into learnable clusters. Subsequently, we guide the model to create identical representations for points having similar soft clustering assignment distributions. This module is trained jointly in an end-to-end manner with the encoder, resembling an EM-style optimization approach in action, where both deep components evolve simultaneously, reflecting a dynamic interplay.

Two theoretical justifications support our proposal: the Laplacian Eigenmap (LE) perspective and the EM viewpoint. Firstly, adopting a spectral manifold learning perspective, the original solely augmentation-based VICReg objective can be formulated as an LE optimization (Balestriero & LeCun, 2022). The implicit augmentation-based relation graph results in disconnected islands corresponding to a rank deficiency of its Laplacian. We mitigate this through explicitly generating the relation graph, facilitating the connection of these islands. Secondly, SSL methods address an underlying two-step EM characterized by two implicit sets of variables (Chen & He, 2021). We distinctly and explicitly outline the representatives of these two steps. The Expectation step (E-step) leverages the learned representation to dynamically generate a relation graph, identifying candidates to promote similar representations. The Maximization step (M-step) refines the encoder by incorporating information from the determined relation graph. These two modules synergize, with the relation graph generator pinpointing candidate pairs for the invariance term and the encoder ensuring the enforcement of identical representation pairs. Our introduced objective function facilitates and stabilizes the joint optimization of these modules, enabling simultaneous execution of the two steps in a single gradient update. Our proposed approach focuses on a node-level representation learning task. Thus, following extensive experimentation across diverse node classification graph datasets, we showcase the superior quality of the learned representation, consistently outperforming previous methods.

## 2    Related Work

### 2.1    Self-supervised Representation Learning

Contrastive SSL (Misra & Maaten, 2020; Bromley et al., 1994; Hjelm et al., 2019; Chen et al., 2020c; Hadsell et al., 2006; Ye et al., 2019; Wu et al., 2018; Chen et al., 2020b) constructs positive pairs employing data augmentation to push their representations closer while pulling apart the negative pairs utilizing InfoNCE (van den Oord et al., 2018). These pairs could be constructed on a mini-batch (Chen et al., 2020a) or employing memory banks (He et al., 2020). Clustering-based SSL (Bautista et al., 2016; Yang et al., 2016; Xie et al., 2016; Huang et al., 2019; Zhuang et al., 2019; Caron et al., 2019; Asano et al., 2019; Yan et al., 2020)

employs a notion of clustering to form the representations by pseudo-labels (Caron et al., 2018) or enforcing identical clustering assignments of augmented pairs instead of directly enforcing identical features (Caron et al., 2020). Knowledge distillation-based SSL leverages student-teacher architectures (Hinton et al., 2015; Grill et al., 2020; Chen & He, 2021; Gidaris et al., 2021; Grill et al., 2020; Gidaris et al., 2020), employing an Exponential Moving Average encoder and a Stop-Gradient (SG) mechanism. Other works maximize the mutual information of representations (Ermolov et al., 2021; Zbontar et al., 2021). Notably, VICReg (Bardes et al., 2021) decorrelates embedding dimensions to prevent preserving redundant information, and VICRegL (Bardes et al., 2022) employs local and global features, enforcing the invariance term for specific image patches.

## 2.2 Graph Representation Learning

GNNs (Kipf & Welling, 2016a; Veličković et al., 2017; Hamilton et al., 2017; Xu et al., 2018; Yang et al., 2021; 2022) accept node features and sufficient annotated data to form representations by recursively aggregating the neighbor's information. Yet, to truly utilize the abundant unlabelled graph data, graph SSL pre-train GNNs to construct node- and graph-level representations without costly labels (García-Durán & Niepert, 2017; Kipf & Welling, 2016b; Bojchevski & Günnemann, 2017). Augmentation-based methods (You et al., 2020; Peng et al., 2020; Hassani & Khasahmadi, 2020; Zhu et al., 2021a;b; 2020c; Thakoor et al., 2021; Zhu et al., 2020b) encourage the representations to be invariant to a specifically designed graph transformation. A BERT-inspired method (Devlin et al., 2018; Hu et al., 2019) masks features of graphs with special structures. DGI (Veličković et al., 2018) aligns a local graph patch with the global one by maximizing the mutual information (Hjelm et al., 2019), followed by edge and node feature extensions (Peng et al., 2020; Jing et al., 2021) and tackling graph classification (Sun et al., 2019). Contrastive methods (Zhu et al., 2020c; 2021b; You et al., 2020; Hassani & Khasahmadi, 2020; Tian et al., 2020; Liu et al., 2023b) are generally inspired by SimCLR (Chen et al., 2020a), employing negative pairs resulting in the sampling bias issue (Bielak et al., 2021), meaning some negative samples may have similar semantics to the anchor while being pulled apart. PGCL (Lin et al., 2022) tackles the sampling bias, constructing the negative pairs from different clusters, while BGRL (Thakoor et al., 2021) adopts the non-contrastive BYOL (Grill et al., 2020). AFGRL (Lee et al., 2022), an augmentation-free BYOL-based method, incorporates kNN, KMeans, and adjacency as local structures and global semantics. Similarly, SPGCL (Wang et al., 2023), an augmentation-free contrastive method, leverages kNN with a single encoder pass.

Prior approaches either heavily rely on augmentation or discard it entirely. We distinguish ourselves by offering a non-contrastive comprehensive framework. In contrast to previous methods, ExGRG incorporates graph augmentations by dynamically adjusting them through the generation and integration of additional guidance in the form of relation graphs. Unlike contrastive SSL, ExGRG avoids issues related to sampling bias and the computational overhead associated with the need for a large number of negative pairs. Our approach constructs an explicit compositional relation graph from various sources through a learnable aggregation mechanism. This enables the identification of candidate pairs for the invariance term and diverges from the traditional use of an implicit augmentation-based relation graph. We stand out by introducing a novel approach leveraging PSEs in an unexplored context. ExGRG marks the pioneering attempt to employ a learnable clustering mechanism to drive the invariance term, departing from relying solely on augmentation.

# 3 Method

## 3.1 Preliminary

### 3.1.1 Graph SSL

As demonstrated in Fig. 1, the initial data points are $M$ nodes within a source graph $\mathcal{G}^s = (\boldsymbol{A}^s, \boldsymbol{X}^s)$, where $\boldsymbol{A}^s \in \mathbb{R}^{M \times M}$ represents the adjacency and $\boldsymbol{X}^s \in \mathbb{R}^{M \times D_{in}}$ signifies the node features. In addition, $\boldsymbol{Y}^s \in \mathbb{R}^M$ denotes the node classification labels. Transformations $\chi_i$ are sampled from the distribution of data augmentations $\chi$ to yield $V$ views as $\mathcal{G}^{(i)} = (\boldsymbol{A}^{(i)}, \boldsymbol{X}^{(i)}) = \chi_i(\mathcal{G}^s)$, where $i \in \{1, \dots, V\}$, $\boldsymbol{A}^{(i)} \in \mathbb{R}^{M \times M}$, $\boldsymbol{X}^{(i)} \in \mathbb{R}^{M \times D_{in}}$, and typically $V = 2$. The mini-batch graph $\mathcal{G} = (\boldsymbol{A}, \boldsymbol{X})$ is then constructed, where $\boldsymbol{A} \in \mathbb{R}^{N \times N}$ is the aggregation of adjacencies $\{\boldsymbol{A}^{(1)}, \dots, \boldsymbol{A}^{(V)}\}$, $\boldsymbol{X} = [\boldsymbol{X}^{(1)}; \dots; \boldsymbol{X}^{(V)}] \in \mathbb{R}^{N \times D_{in}}$, and $N =$

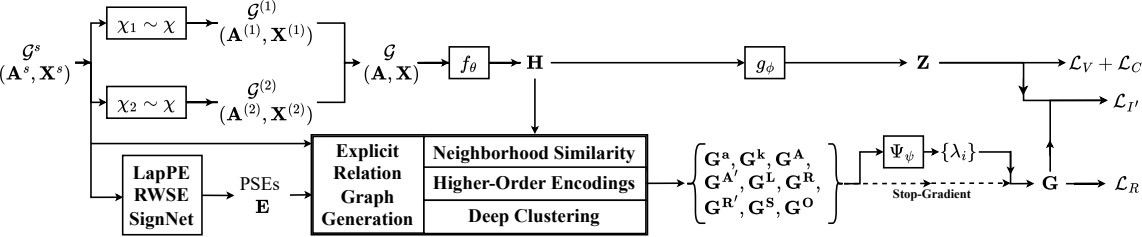

Figure 1: The comprehensive architecture of ExGRG. We incorporate augmented views $\mathcal{G}^{(1)}$ and $\mathcal{G}^{(2)}$ from source graph $\mathcal{G}^{(s)}$ to compose an input graph $\mathcal{G}$, which is then fed into encoder $f_\theta$ and expander $g_\phi$ to produce representations $\boldsymbol{H}$ and embeddings $\boldsymbol{Z}$. We explicitly generate a compositional relation graph $\boldsymbol{G}$, aggregated from various intermediate relation graphs, to guide the invariance term $\mathcal{L}_{I'}$ instead of relying solely on augmentations.

$VM = 2M$ denotes the mini-batch size. This graph $\mathcal{G}$ is subsequently fed into the graph encoder $f_\theta$, which is usually a GNN. The representations $\boldsymbol{H} = f_\theta(\mathcal{G}) \in \mathbb{R}^{N \times D_H}$ produce embeddings $\boldsymbol{Z} = g_\phi(\boldsymbol{H}) \in \mathbb{R}^{N \times D_Z}$, employing the expander $g_\phi$, interchangeably referred to as the projector or decoder in the literature, which is a Multi-Layer Perceptron (MLP).

The essence of SSL lies in employing signals inherent in data points as a form of supervision. SSL pre-training aims to discover representations of input points by identifying $f_\theta$ and $g_\phi$. Subsequently, the pre-trained $f_\theta$ is utilized in downstream tasks, predicting $\boldsymbol{Y}^s$ from $\mathcal{G}^s$, through linear probing, nonlinear probing, or fine-tuning. In linear probing, the prevailing approach in graph SSL, a single linear layer is affixed to the frozen pre-trained $f_\theta$ while detaching $g_\phi$. This linear layer is trained for the downstream node classification task, utilizing the frozen representations as inputs. Though ExGRG centers around a node-level representation learning task, it can be seamlessly extended to learn graph- and link-level representations, with details provided in *Appendix* § A.1.

### 3.1.2 VICReg and Relation Graph

Contrastive SSL prevents the representation collapse by incorporating negative pairs (Chen et al., 2020a). In contrast, the VICReg objective $\mathcal{L}_{Vic}$ (Eq. 2) introduces constraints to maximize the volume of the embedding space through two regularization terms, variance $\mathcal{L}_V$ and covariance $\mathcal{L}_C$ (Eq. 1). The term $\mathcal{L}_V$ ensures variance of embedding dimensions exceeds a threshold, while $\mathcal{L}_C$ enforces decorrelation between every pair of dimensions. The invariance term $\mathcal{L}_I$ (Eq. 1) encourages similar representations for semantically related points through augmentation.

$$\mathcal{L}_V = \sum_{k=1}^{D_Z} \max\left(0, 1 - \sqrt{\mathrm{Cov}(\boldsymbol{Z})_{k,k}}\right), \quad \mathcal{L}_C = \sum_{k=1}^{D_Z} \sum_{j=1, j\neq k}^{D_Z} \mathrm{Cov}(\boldsymbol{Z})_{k,j}^2, \quad \mathcal{L}_I = \sum_{(i,j)\in S_a} \|\boldsymbol{Z}_{i,.} - \boldsymbol{Z}_{j,.}\|_2^2. \quad (1)$$

$$\mathcal{L}_{Vic} = \alpha\mathcal{L}_V + \beta\mathcal{L}_C + \gamma\mathcal{L}_I \quad (2)$$

The invariance term $\mathcal{L}_I$ is exclusively guided by augmentations, performed on data points with index pairs $(i,j)$ in an implicitly defined set $S_a = \{(i,j) \in \mathbb{N}^2 \mid \boldsymbol{X}_{i,.} \text{ and } \boldsymbol{X}_{j,.} \text{ are augmented from same source } \boldsymbol{X}_{k,.}^s \text{ s.t. } 1 \leq i < j \leq N \text{ and } 1 \leq k \leq \frac{N}{2}\}$. The notation $\boldsymbol{X}$ encompasses all points from both views within a unified matrix. Thus, with $(i,j) \in S_a$, the $i$-th point $\boldsymbol{X}_{i,.}$ is the same as the $k$-th point in the first view, denoted as $\boldsymbol{X}_{k,.}^{(1)}$. Similarly, the $j$-th point $\boldsymbol{X}_{j,.}$ is the same as the $k$-th point in the second view, represented as $\boldsymbol{X}_{k,.}^{(2)}$. The pair $(\boldsymbol{X}_{i,.}, \boldsymbol{X}_{j,.})$ is augmented from the same source point $\boldsymbol{X}_{k,.}^s$. In the original invariance term $\mathcal{L}_I$, the enforcement is applied only over $(i,j) \in S_a$. We can conceptualize the relations among data points determined by this augmentation using a relation matrix $\boldsymbol{G}^a \in \mathbb{R}^{N \times N}$, where $\boldsymbol{G}_{i,j}^a = 1$ if $(i,j) \in S_a$, and $\boldsymbol{G}_{i,j}^a = 0$ otherwise. A relation matrix can be interpreted as an adjacency matrix for a relation graph, though we may use the terms *relation graph* and *relation matrix* interchangeably.

### 3.2 Explicit Compositional Relation Graph

Eq. 3 demonstrates the ExGRG's invariance term $\mathcal{L}_{I'}$ employing our explicitly-generated compositional relation graph $\boldsymbol{G} \in \mathbb{R}^{N \times N}$ as opposed to only relying on augmentations $\boldsymbol{G}^a$. Our invariance term $\mathcal{L}_{I'}$ may be reduced to the original $\mathcal{L}_I$ by simply setting $\boldsymbol{G} = \boldsymbol{G}^a$.

$$\mathcal{L}_{I'} = \sum_{i=1}^{N} \sum_{j=1}^{N} \boldsymbol{G}_{i,j} \|\boldsymbol{Z}_{i,.} - \boldsymbol{Z}_{j,.}\|_2^2 \tag{3}$$

Two theoretical justifications underpin the rationale behind explicitly generating $\boldsymbol{G}$. This is motivated by the interplay between SSL with (i) the LE method and (ii) the EM optimization.

#### 3.2.1 Laplacian Eigenmap Perspective

In variance-covariance-constrained VICReg, $\alpha = \beta \gg \gamma$, signifying that the variance and covariance terms in $\mathcal{L}_{Vic}$ are already satisfied (Eq. 2). Optimizing this VICReg objective is equivalent to solving the LE problem in *Appendix* Eq. 11 (Balestriero & LeCun, 2022). The distinction lies in VICReg's $\boldsymbol{G}$ being determined through data augmentations, while in LE, it can be designed by forming kNN graphs in the input. Due to the duality between $S_a$ and $G^a$, the augmentation-based relation graph $G^a$ consists of $M = \frac{N}{2}$ disconnected islands. Therefore, the corresponding Laplacian $L^a$ for $\boldsymbol{G}^a$ suffers from rank deficiency, as it has $\frac{N}{2}$ zero eigenvalues. In addition, due to the shortcomings of graph augmentations, connections between pairs of data points within each island in $\boldsymbol{G}^a$ tend to be much less valuable compared to the vision domain. To address this challenge, we mitigate the rank deficiency of $L^a$ by constructing a more informative $\boldsymbol{G}$ through the incorporation of additional sources to guide the invariance term. We introduce meaningful entries to $\boldsymbol{G}$, establishing connections among the disconnected islands while assigning soft importance to each connection.

#### 3.2.2 Alternating Two-Step EM Perspective

SimSiam (Chen & He, 2021) hypothesizes that SSL methods with Siamese architectures share an underlying optimization problem that could be effectively modeled employing the EM framework. In this optimization, SSL pre-training deals with two implicit sets of variables, reflecting an implied alternating optimization for each set in EM. Employing SG emerges as a viable approach to prevent collapse, resembling the alternating two-step EM optimization style in a single gradient update. The E-step is performed utilizing a predictor. Instead of directly comparing augmented views, the predictor assigns implicit targets for each anchor point, upon which the invariance loss is applied. In the M-step, the encoder is updated but only through one of the two available paths employing the SG technique.

ExGRG departs from computing these implicit targets. Instead, we opt for an explicit, end-to-end methodology that leverages nodes' intrinsic information and characteristics to construct the relation graph. The connections in our relation graph determine candidate targets for each anchor point, formulated as the entries within $\boldsymbol{G}$. Within our framework, the two EM steps are consolidated into a single update via a joint optimization. Notably, ExGRG diverges from the implicit sets mentioned earlier. Instead, we utilize distinct and explicit sets of parameters. Specifically, leveraging our proposed loss function, both the parameters of the encoder and the relation graph generation modules are simultaneously updated.

In the EM algorithm, the E-step assigns data points to different given distributions. In ExGRG, this assignment corresponds to constructing $\boldsymbol{G}$, indicating which points should be brought closer. This E-step offers a roadmap for our M-step, which incorporates the invariance term $\mathcal{L}_{I'}$ based on the constructed $\boldsymbol{G}$. Optimizing $\mathcal{L}_{I'}$ leads to updating $f_\theta$ and $g_\phi$, leading to new $\boldsymbol{H}$ and $\boldsymbol{Z}$, which subsequently determine a new $\boldsymbol{G}$ in the E-step of the next iteration.

#### 3.2.3 Intra-view and Soft Relations

An invariance term $\mathcal{L}_{I'}$ based on constructing $\boldsymbol{G}$ enables us to capture and enforce similarities among points within a view (intra-view) or across different views (inter-view). In essence, the enforcement of invariance criteria occurs concurrently for an anchor point in three key scenarios: **(i)** with its corresponding point in

the other view, both resulting from augmentations on the same source, **(ii)** with its analogous points from the same view representing augmentations from different source points but sharing similar semantics to the anchor point, and **(iii)** with some analogous points from the other view holding similar characteristics as described in (ii). While conventional methods only support scenario (i), our extension enables us to integrate additional information from the same mini-batch. Further details can be found in *Appendix* § B.5.1.

Another distinction in our approach is the incorporation of support for soft values of relations. Unlike the binary nature of the conventional $\boldsymbol{G}^a$ entries (0 or 1), we allow for soft values that indicate the degree of enforcing identical representations for a candidate pair in $\boldsymbol{G}$. This introduces flexibility and indicates the relative importance of each connection pair compared to other pairs resembling LE.

### 3.3 Strategies for Constructing Relation Graphs

Our approach to enhancing the invariance term involves exploring ways to construct more informative relation graphs. This empowers the invariance term to leverage additional similarities beyond augmentations through **(i)** utilizing *Neighborhood Similarity* employing the kNN graph, **(ii)** exploiting *Higher-Order Encodings* as the structural and positional similarities of nodes, and **(iii)** incorporating an end-to-end *Deep Clustering* algorithm.

#### 3.3.1 Neighborhood Similarity in the Representation Space

We can explicitly determine a relation graph by leveraging the similarities between the representations of points. This approach encourages the convergence of embeddings for points with similar neighborhoods in the representation space, resembling LE. Thus, we compare the points in $\boldsymbol{H}$ to compute the kNN relation matrix denoted by $\boldsymbol{G}^k = F_k(\boldsymbol{G}^H) \in \mathbb{R}^{N \times N}$, where $\boldsymbol{G}^H_{i,j} = sim(\boldsymbol{H}_{i,.}, \boldsymbol{H}_{j,.})$ and $i, j \in \{1, \ldots, N\}$. The function $sim$ represents a notion of similarity, such as Euclidean or cosine similarity. Additionally, $F_k$ sets entries other than the $k$-largest values in each row of its input to zero, resembling the kNN algorithm.

Our empirical analysis indicates that focusing solely on inter-view relations in $\boldsymbol{G}^k$ yields performance comparable to a scenario that also includes intra-view relations, while significantly reducing computational costs. We attribute this to the sufficiently strong relations identified by k-nearest neighbors from the alternate view. Further details are provided in § 4.3 and *Appendix* § B.5.1.

#### 3.3.2 Higher-Order Graph Encodings

In homophilic input graphs, the proximity of two nodes in $\mathcal{G}^s$ indicates sharing the same class label in $\boldsymbol{Y}^s$ (Zhu et al., 2020a). Therefore, depending on the degree of homophily in the source graph $\mathcal{G}^s$, the edges between source points in $\boldsymbol{A}^s$ can be exploited to construct an informative relation matrix $\boldsymbol{G}^A = F_A(\boldsymbol{A}^s) \in \mathbb{R}^{N \times N}$, where $F_A$ extracts the adjacency information corresponding to the $N$ points in the mini-batch. Recognizing that $\boldsymbol{G}^k$ or $\boldsymbol{G}^A$ are often noisy on their own (Lee et al., 2022), we propose another more conservative filtered version through element-wise multiplication as $\boldsymbol{G}^{A'} = \boldsymbol{G}^A \odot \boldsymbol{G}^k$. This $\boldsymbol{G}^{A'}$ retains the soft similarities of $\boldsymbol{G}^k$, with additional details provided in *Appendix* § A.5.

Additionally, drawing inspiration from LE, we can conduct pre-processing on the input to extract characteristics that can be utilized to construct a more informative $\boldsymbol{G}$. This could involve leveraging the comparison of some statistics and features at the level of individual nodes, neighborhoods, or sub-graphs. To achieve this, we incorporate positional or structural similarities of nodes. Our approach differs from their typical application in transformers to enhance the identifiability of nodes within a graph (Liu et al., 2023a). To the best of our knowledge, this is the first attempt to integrate PSEs into the SSL pre-training.

Positional Encodings (PEs) attempt to encode the position of a node within a graph. Thus, comparing these encodings can help identify nodes with similar positions in the graph, capturing a higher-level understanding of what the adjacency matrix can signify. Alternatively, Structural Encodings (SEs) capture the structural characteristics of the neighboring region surrounding each node. By incorporating SEs, ExGRG gains access to rich information about nodes with analogous local and global connectivity patterns. Our approach encompasses three PSEs: Laplacian Eigenvectors Positional Encoding (LapPE) (Dwivedi et al., 2023), Random Walk Structural Encodings (RWSE) (Dwivedi et al., 2021), and SignNet (Lim et al., 2022).

LapPE serves as a global PE, offering insights into the node's overall position within the graph. RWSE provides local structural information, indicating the sub-structure to which a node belongs. SignNet, acting as another global positional encoder, is also capable of capturing some local structural information.

To construct a relation matrix from these PSEs, a comparison of encodings for pairs of nodes is necessary, followed by the formation of candidate pairs based on a construction algorithm such as kNN. By denoting a computed PSE as $\boldsymbol{E} \in \mathbb{R}^{N \times D_E}$, we obtain $\boldsymbol{G}^{PSE} = F_k(\boldsymbol{G}^E) \in \mathbb{R}^{N \times N}$, where $\boldsymbol{G}^E_{i,j} = sim(\boldsymbol{E}_{i,.}, \boldsymbol{E}_{j,.})$ and $i, j \in \{1, \ldots, N\}$. By incorporating LapPE, RWSE, and SignNet PSEs into the $\boldsymbol{G}^{PSE}$ placeholder, we construct three relation matrices, $\boldsymbol{G}^L$, $\boldsymbol{G}^R$, and $\boldsymbol{G}^S$. Similar to $\boldsymbol{G}^{A'}$, we observe the filtered RWSE $\boldsymbol{G}^{R'} = \boldsymbol{G}^R \odot \boldsymbol{G}^k$ to be more beneficial for certain datasets.

### 3.3.3 Deep Clustering for Relation Graph Generation

Various methods have been introduced to jointly optimize a deep feature extractor module and a deep clustering module (Zhou et al., 2022). Conventionally, this is performed by leveraging the feature extractor module to provide essential features for the clustering module, with the aim of addressing *a clustering task*. However, in our approach, we invert this paradigm by proposing a novel relation between these two modules through our introduced objective function. Instead, we leverage the clustering module to explicitly guide the feature extractor module towards *a representation learning task*.

Our encoder and learnable clustering modules collaboratively aim to enhance the representations. The clustering module employs learnable prototypes to assign a probability distribution to each data point. This distribution indicates the confidence of assigning a data point to different clusters. Subsequently, these assignments are employed to guide the invariance loss. This encourages the encoder to map points with similar clustering assignments to identical representations. Both the prototypes and the encoder are learned jointly, providing a more integrated and effective approach.

This marks a novel attempt to integrate a learnable clustering algorithm for representation learning, inspired by a non-learnable clustering approach, KMeans (Lee et al., 2022). While a clustering algorithm is previously employed in SSL methods to partition the points (Caron et al., 2020), a key distinction lies in the representation learning aspect. In these methods, the invariance loss is still guided through data augmentation. This means that for points in $S_a$, instead of ensuring identical features directly through $\mathcal{L}_I$, they aim for identical clustering assignments. In contrast, ExGRG positions the clustering module as an explicit and direct guide, promoting the same representation for data points with similar cluster assignments at each iteration. In essence, our proposal deviates from the approach of enforcing identical cluster assignments for augmentation-based paired points; instead, we enforce identical representations for points with similar cluster assignments. The candidate pairs for this enforcement encompass the entire mini-batch, not only pairs augmented from the same source.

In our deep clustering module, we leverage optimal transport (Asano et al., 2019) to partition the points employing $K$ trainable prototypes $\boldsymbol{C} \in \mathbb{R}^{K \times D_H}$. The probabilities $\boldsymbol{P} \in \mathbb{R}^{N \times K}$, represent the distribution of assignments of points to different clusters. They are computed by comparing the representations $\boldsymbol{H}$ with prototypes $\boldsymbol{C}$, followed by a softmax function with temperature $\tau$. The probability $\boldsymbol{P}_{j,k}$ of assigning the point $\boldsymbol{H}_{j,.} \in \mathbb{R}^{D_H}$ to the $k$-th prototype $\boldsymbol{C}_{k,.} \in \mathbb{R}^{D_H}$ is given by

$$\boldsymbol{P}_{j,k} = \frac{\exp(\frac{1}{\tau}\boldsymbol{H}_{j,.}^T \boldsymbol{C}_{k,.})}{\sum_{k'} \exp(\frac{1}{\tau}\boldsymbol{H}_{j,.}^T \boldsymbol{C}_{k',.})}. \tag{4}$$

To update $\boldsymbol{P}$, we aim to make them more similar to codes $\boldsymbol{Q} \in \mathbb{R}^{N \times K}$, which serve as a refined version of $\boldsymbol{P}$ computed from the optimal transport problem. The underlying concept of the target probability distribution $\boldsymbol{Q}$ is evenly distributing data points among clusters. The updates for $\boldsymbol{P}$ and $\boldsymbol{C}$ are performed leveraging our loss term $\mathcal{L}_O$ that aligns $\boldsymbol{P}$ and $\boldsymbol{Q}$ using cross-entropy for all pairs in $S_O$ as

$$\mathcal{L}_O = \frac{1}{|S_O|} \sum_{(i,j) \in S_O} l_{i,j}, \quad \text{where } l_{i,j} = -\sum_k \boldsymbol{Q}_{i,k} \log \boldsymbol{P}_{j,k}, \quad \text{and } S_O = \{(i,j) \in \mathbb{N}^2 \mid i = j\} \cup S_a. \tag{5}$$

At each iteration, $\boldsymbol{Q}$ is derived directly from $\boldsymbol{C}$ and $\boldsymbol{H}$ employing the Sinkhorn-Knopp algorithm (Cuturi, 2013; Asano et al., 2019; Caron et al., 2020), with further details provided in *Appendix* § A.6. Simultaneously, $\boldsymbol{P}$ is aligned with $\boldsymbol{Q}$ to adjust $\boldsymbol{C}$ accordingly for the subsequent iterations. To incorporate the auxiliary online extracted information available in $\boldsymbol{P}$, a relation matrix $\boldsymbol{G}^O \in \mathbb{R}^{N \times N}$ is constructed as

$$\boldsymbol{G}^O = F_K \left( F_n(\boldsymbol{G}^P) \right), \quad \text{where } \boldsymbol{G}^P_{i,j} = \sum_k \boldsymbol{P}_{i,k} \log \boldsymbol{P}_{j,k}. \tag{6}$$

Here, $F_n$ normalizes entries of $\boldsymbol{G}^P$ to $[0,1]$. Additionally, $F_K$ selects the top $K_g$ entries of the input matrix globally and sets the rest to zero. As a result, $\boldsymbol{G}^O$ is a sparse normalized version of $\boldsymbol{G}^P$. The entry $\boldsymbol{G}^P_{i,j}$ represents negative cross-entropy over distributions $\boldsymbol{P}_{i,.}$ and $\boldsymbol{P}_{j,.}$, indicating a notion of similarity for clustering assignments. If points $\boldsymbol{H}_{i,.}$ and $\boldsymbol{H}_{j,.}$ share high similarity in their clustering assignments, signifying similar characteristics, their representations are encouraged to be closer through $\boldsymbol{G}^O$.

### 3.4 Multi-Source Aggregation

To compute the invariance term $\mathcal{L}_{I'}$ (Eq. 3), we construct a compositional relation matrix $\boldsymbol{G}$ through aggregating various relation matrices $\boldsymbol{G}^{(i)} \in S_G$ derived from previously-discussed sources gathered in $S_G$ (Eq. 7). In a straightforward scenario, this aggregation can be accomplished through summation with learnable coefficients $\lambda_i$. This is illustrated in Eq. 7, where $F_{SG}$, outlined in § 3.5, enforces the Stop-Gradient (SG) mechanism. During the forward pass, $F_{SG}$ acts as the identity function while ensuring that no gradients are propagated during the backward pass. Combining Eq. 7 and Eq. 3, we observe that this effective aggregation can also be represented as individual invariance terms corresponding to each of the $\boldsymbol{G}^{(i)} \in S_G$. Besides, employing a softmax, we ensure $\lambda_i$ are normalized, guaranteeing that they collectively sum up to one, i.e., we enforce $\sum_{\boldsymbol{G}^{(i)} \in S_G} \lambda_i = 1$.

$$\boldsymbol{G} = \sum_{\boldsymbol{G}^{(i)} \in S_G} \lambda_i F_{SG}(\boldsymbol{G}^{(i)}), \quad \text{where } S_G = \{\boldsymbol{G}^a, \boldsymbol{G}^k, \boldsymbol{G}^A, \boldsymbol{G}^{A'}, \boldsymbol{G}^L, \boldsymbol{G}^R, \boldsymbol{G}^{R'}, \boldsymbol{G}^S, \boldsymbol{G}^O\}. \tag{7}$$

To generate the learnable coefficients $\lambda_i$, we employ an MLP $\Psi_\psi$ as a hypernetwork (Ha et al., 2016). This aims to efficiently aggregate an arbitrary number of relation graphs using a single module in an online manner. We leverage a hypernetwork-inspired formulation for its computational capabilities encompassing information-sharing, compressed nature, and expedited training process (Chauhan et al., 2023). This module receives some characteristics about each $\boldsymbol{G}^{(i)} \in S_G$ in an online manner as

$$\lambda_i = \Psi_\psi \left( F_s(\boldsymbol{G}^{(i)}) \right), \tag{8}$$

where $F_s$ denotes a function that computes two simple yet effective statistics from its input $\boldsymbol{G}^{(i)}$ to determine the appropriate contribution to $\boldsymbol{G}$: (i) the sum of $\boldsymbol{G}^{(i)}$ entries, reflecting the average strength of connections within $\boldsymbol{G}^{(i)}$, and (ii) the count of $\boldsymbol{G}^{(i)}$ non-zero entries, resembling the level of sparsity.

### 3.5 End-to-End Training Procedure

The transition between optimizing our modules, specifically the alteration of (i) updating $f_\theta$ and $g_\phi$, and (ii) generation of a more informative $\boldsymbol{G}^{(i)} \in S_G$ and eventually $\boldsymbol{G}$, can be achieved through the incorporation of an SG mechanism (Eq. 7), inspired by Grill et al. (2020) and Chen & He (2021), alongside a relation matrix regularization as

$$\mathcal{L}_R = -\sum_{i,j} (\boldsymbol{G}_{i,j})^2. \tag{9}$$

These two components are introduced to prevent $\boldsymbol{G}^{(i)}$ and $\boldsymbol{G}$ from collapsing to a degenerate solution, wherein all entries are encouraged to be zero under the influence of $\mathcal{L}_{I'}$. This integration enables us to squeeze the two EM steps into a single gradient update, thus enabling joint optimization. This strategy aims

Table 1: Downstream performance measured in terms of node classification accuracy's mean and standard deviation over 20 random model initializations and dataset splits. *OOM* indicates Out Of Memory.

| Model | WikiCS | AmzComp | AmzPhoto | CoCS | CoPhy | Cora | CiteSeer | PubMed | DBLP |
|---|---|---|---|---|---|---|---|---|---|
| Supervised MLP | $71.98 \pm 0.42$ | $73.81 \pm 0.21$ | $78.53 \pm 0.32$ | $90.37 \pm 0.19$ | $93.58 \pm 0.41$ | $47.92 \pm 0.41$ | $49.31 \pm 0.26$ | $69.14 \pm 0.34$ | - |
| Supervised GCN | $77.19 \pm 0.12$ | $86.51 \pm 0.54$ | $92.42 \pm 0.22$ | $93.03 \pm 0.31$ | $95.65 \pm 0.16$ | $81.54 \pm 0.68$ | $70.73 \pm 0.65$ | $79.16 \pm 0.25$ | $82.7 \pm 0.00$ |
| Node2Vec | $71.79 \pm 0.05$ | $84.39 \pm 0.08$ | $89.67 \pm 0.12$ | $85.08 \pm 0.03$ | $91.19 \pm 0.04$ | $71.08 \pm 0.91$ | $47.34 \pm 0.84$ | $66.23 \pm 0.95$ | - |
| DeepWalk | $74.35 \pm 0.06$ | $85.68 \pm 0.06$ | $89.44 \pm 0.11$ | $84.61 \pm 0.22$ | $91.77 \pm 0.15$ | $70.72 \pm 0.63$ | $51.39 \pm 0.41$ | $73.27 \pm 0.86$ | - |
| DW + Features | $77.21 \pm 0.03$ | $86.28 \pm 0.07$ | $90.05 \pm 0.08$ | $87.70 \pm 0.04$ | $94.90 \pm 0.09$ | - | - | - | - |
| DGI | $75.35 \pm 0.14$ | $83.95 \pm 0.47$ | $91.61 \pm 0.22$ | $92.15 \pm 0.63$ | $94.51 \pm 0.52$ | $82.34 \pm 0.71$ | $71.83 \pm 0.54$ | $76.78 \pm 0.31$ | - |
| GMI | $74.85 \pm 0.08$ | $82.21 \pm 0.31$ | $90.68 \pm 0.17$ | OOM | OOM | $82.39 \pm 0.65$ | $71.72 \pm 0.15$ | $79.34 \pm 1.04$ | - |
| MVGRL | $77.52 \pm 0.08$ | $87.52 \pm 0.11$ | $91.74 \pm 0.07$ | $92.11 \pm 0.12$ | $95.33 \pm 0.03$ | $83.45 \pm 0.68$ | $73.28 \pm 0.48$ | $80.09 \pm 0.62$ | - |
| GRACE | $77.97 \pm 0.63$ | $86.50 \pm 0.33$ | $92.46 \pm 0.18$ | $92.17 \pm 0.04$ | OOM | $81.92 \pm 0.89$ | $71.21 \pm 0.64$ | $80.54 \pm 0.36$ | - |
| GCA | $77.94 \pm 0.67$ | $87.32 \pm 0.50$ | $92.39 \pm 0.33$ | $92.84 \pm 0.15$ | OOM | $82.07 \pm 0.10$ | $71.33 \pm 0.37$ | $80.21 \pm 0.39$ | - |
| BGRL | $79.98 \pm 0.10$ | $90.34 \pm 0.19$ | $93.17 \pm 0.30$ | $93.31 \pm 0.13$ | $95.73 \pm 0.05$ | $83.83 \pm 1.61$ | $72.32 \pm 0.89$ | $86.03 \pm 0.33$ | $84.07 \pm 0.23$ |
| AFGRL | $77.62 \pm 0.49$ | $89.88 \pm 0.33$ | $93.22 \pm 0.28$ | $93.27 \pm 0.17$ | $95.69 \pm 0.10$ | $81.60 \pm 0.54$ | $71.02 \pm 0.37$ | $80.02 \pm 0.48$ | - |
| SPGCL | $79.01 \pm 0.51$ | $89.68 \pm 0.19$ | $92.49 \pm 0.31$ | $91.92 \pm 0.10$ | $95.12 \pm 0.15$ | $83.16 \pm 0.13$ | $71.96 \pm 0.42$ | $79.16 \pm 0.73$ | - |
| ExGRG | $\mathbf{82.09 \pm 0.67}$ | $\mathbf{93.37 \pm 0.48}$ | $\mathbf{96.42 \pm 0.54}$ | $\mathbf{94.57 \pm 0.31}$ | $\mathbf{96.59 \pm 0.20}$ | $\mathbf{97.87 \pm 0.55}$ | $\mathbf{89.68 \pm 1.46}$ | $\mathbf{88.03 \pm 0.49}$ | $\mathbf{86.01 \pm 0.59}$ |

to enhance the synergy between the encoder's parameter adjustment and the refinement of the informative content within the relation matrices. Additional details are provided in *Appendix* § A.7.

Finally, to facilitate end-to-end training of our model through gradient descent, we employ a comprehensive multi-term loss function $\mathcal{L}_{ETE}$ that encompasses the terms discussed so far as

$$\mathcal{L}_{ETE} = \alpha \mathcal{L}_V + \beta \mathcal{L}_C + \gamma \mathcal{L}_{I'} + \alpha_1 \mathcal{L}_O + \alpha_2 \mathcal{L}_R, \tag{10}$$

where $\alpha$, $\beta$, and $\gamma$, $\alpha_1$ and $\alpha_2$ serve as coefficients for the respective loss terms.

## 4 Experimental Evaluation

### 4.1 Experimental Setup

We undertake an extensive experimental analysis, aiming to demonstrate the efficacy of our approach and its superiority in comparison to other state-of-the-art methods. A GCN (Kipf & Welling, 2016a) as the encoder $f_\theta$ is utilized, with details provided in *Appendix* § B.2. Also, we follow the linear probing protocol, the established approach in prior graph SSL works (Thakoor et al., 2021). Further details can be found in § 3.1.1 and *Appendix* § B.3.

**Datasets:** We employ a wide range of real-world graphs within 9 node classification datasets, including WikiCS, Amazon Computers (AmzComp), Amazon Photo (AmzPhoto), Coauthor CS (CoCS), Coauthor Physics (CoPhy), Cora, CiteSeer, PubMed, and DBLP, with corresponding statistics in *Appendix* Table 3.

**Baselines:** We compare ExGRG with various graph representation learning methods, encompassing (i) supervised MLP employing raw node features and GCN (Kipf & Welling, 2016a), (ii) conventional unsupervised graph embedding works Node2Vec (Grover & Leskovec, 2016) and DeepWalk (Perozzi et al., 2014), alongside (iii) state-of-the-art contrastive and non-contrastive SSL methods, including DGI (Veličković et al., 2018), GMI (Peng et al., 2020), MVGRL (Hassani & Khasahmadi, 2020), GRACE (Zhu et al., 2020c), GCA (Zhu et al., 2021b), BGRL (Thakoor et al., 2021), and augmentation-free works AFGRL (Lee et al., 2022) and SPGCL (Wang et al., 2023).

### 4.2 Experimental Results

The empirical performance compared to the baselines is presented in Table 1, showcasing node classification accuracy's mean and standard deviation over 20 trials. Each trial corresponds to a random model initialization and distinct train-validation-test splits. Data for other methods are sourced from previous works where available (Thakoor et al., 2021; Lee et al., 2022; Wang et al., 2023). ExGRG consistently outperforms all baselines across these datasets. This demonstrates the effectiveness of our proposed framework in explicitly generating relation graphs by incorporating various forms of prior knowledge, such as PSEs and adjacency,

Table 2: Ablation studies on Amazon Computers. Models are altered concerning ExGRG (5k iters) as the reference. The metrics are outlined in *Appendix* § B.4.

| [Ablated] Model | Accuracy | corr $H$ | corr $Z$ | std $H$ | std $Z$ | nstd $H$ | rank $H$ | rank $Z$ |
|---|---|---|---|---|---|---|---|---|
| **ExGRG (9k iters)** | 93.37 ± 0.48 | 0.0025 | 0.1072 | 0.1536 | 0.9185 | 0.0447 | 512 | 1024 |
| **ExGRG (5k iters)** | 93.26 ± 0.57 | **0.0009** | 0.0866 | 0.1250 | **0.6769** | 0.0448 | **512** | **1024** |
| **Binary $G$** | 93.21 ± 0.40 | **142.28** | 0.0012 | 0.7460 | **0.0985** | 0.0413 | **279** | **777** |
| **No $\mathcal{L}_R$** | 92.89 ± 0.50 | **5513.5** | 0.0001 | 1.5969 | **0.0222** | 0.0378 | **19** | **272** |
| **No $\Psi_\psi$** | 92.90 ± 0.52 | **5514.3** | 0.0001 | 1.5971 | **0.0223** | 0.0378 | **19** | **269** |
| **No $F_{SG}$ on $G^S$** | 93.12 ± 0.51 | 0.0009 | 0.0825 | 0.1200 | 0.6726 | 0.0448 | 512 | 1024 |
| **Add Standalone $G^k$** | 91.67 ± 0.47 | 0.0006 | 0.0485 | 0.1127 | **0.4813** | 0.0439 | 512 | 1024 |
| **No $G^{A'}$ or $G^{R'}$** | 93.11 ± 0.47 | 0.0010 | 0.0544 | 0.1200 | **0.5708** | 0.0448 | 512 | 1024 |
| **No $G^a$** | 93.03 ± 0.52 | 0.0010 | 0.0550 | 0.1202 | **0.5768** | 0.0448 | 512 | 1024 |
| **No $G^{A'}$** | 93.03 ± 0.53 | 0.0010 | 0.0550 | 0.1202 | **0.5768** | 0.0448 | 512 | 1024 |
| **No $G^L$** | 92.96 ± 0.58 | 0.0007 | 0.0778 | 0.1161 | 0.6552 | 0.0448 | 512 | 1024 |
| **No $G^{R'}$** | 92.94 ± 0.47 | 4348.2 | 0.0001 | 1.5292 | **0.0224** | 0.0374 | **23** | **294** |
| **$G^R$ instead of $G^{R'}$** | 92.89 ± 0.61 | 0.0043 | 0.0602 | 0.1326 | **0.4390** | 0.0446 | 512 | 1024 |
| **No $G^S$** | 93.15 ± 0.53 | 0.0011 | 0.0729 | 0.1227 | 0.6525 | 0.0448 | 512 | 1024 |
| **No $G^{PSE}$ or $G^{A'}$** | 93.04 ± 0.49 | 0.0007 | 0.0780 | 0.1167 | 0.6568 | 0.0448 | 512 | 1024 |
| **No $G^O$** | 92.82 ± 0.56 | 4519.4 | 0.0001 | 1.5336 | **0.0217** | 0.0346 | **22** | **297** |
| $F_k$ instead of $F_K$ in $G^O$ | 93.12 ± 0.51 | 0.0009 | 0.0825 | 0.1200 | 0.6726 | 0.0448 | 512 | 1024 |
| $S_O = \{(i,j)\mid i=j\}$ | 93.12 ± 0.51 | 0.0009 | 0.0825 | 0.1200 | 0.6726 | 0.0447 | 512 | 1024 |
| $S_O = S_a$ | 93.12 ± 0.51 | 0.0009 | 0.0825 | 0.1200 | 0.6726 | 0.0448 | 512 | 1024 |
| **Intra Relations** | 93.26 ± 0.54 | 0.0027 | 0.1064 | 0.1524 | **0.9322** | **0.0321** | 512 | 1024 |

as well as online extracted information through a kNN graph and a deep clustering module. Concerning Table 1, the following insights are noteworthy.

**(i)** Previous SSL methods consistently outperform supervised GCN with a notable gap across all datasets except the large-scale ones CoCS and CoPhy, where the top-performing method, BGRL, achieves comparable results with GCN. However, ExGRG maintains superiority even under this scenario. **(ii)** When comparing MLP with GCN, the performance gap is more pronounced in Cora and CiteSeer compared to CoCS and CoPhy, underscoring the importance of graph structural properties facilitated by the message-passing mechanism. This trend aligns with the datasets where the ExGRG's most significant improvements are observed compared to the state-of-the-art. This indicates the effectiveness of incorporating adjacency and PSEs into the learning process through their corresponding relation graphs. **(iii)** Despite the limitations of graph augmentations, comparing the two BYOL-based methods, BGRL with the augmentation-free AFGRL across WiKiCS, AmzComp, Cora, CiteSeer, and PubMed, we observe the superiority of employing graph augmentations. This validates our approach of not solely relying on augmentations while still utilizing them in $G^a$ alongside other informative sources.

**(iv)** The significance of our non-contrastive approach is highlighted in PubMed, where BGRL, the other non-contrastive work, also outperforms contrastive approaches GRACE, GCA, and SPGCL by a considerable margin. We hypothesize that contrastive methods suffer from the sampling bias issue in such scenarios, where the representation of every other point is treated as negative samples to be pushed apart from the anchor. In contrast, we adopt volume maximization terms $\mathcal{L}_V$ and $\mathcal{L}_C$ alongside explicitly determining which points should be connected in our compositional relation graph, leading to identical representations in $\mathcal{L}_{I'}$. **(v)** Contrastive approaches also face significant memory consumption due to the excessive number of negative pairs, making them incapable of handling large-scale datasets like CoCS and CoPhy. Conversely, our non-contrastive approach manages to outperform even with a small mini-batch size $N$, as demonstrated in *Appendix* Fig. 10.

## 4.3 Ablation Studies

To showcase the effectiveness of our design choices and underscore the significance of each introduced component and loss term in $\mathcal{L}_{ETE}$, we conduct ablation studies on AmzComp (Table 2), alongside CiteSeer and Cora (*Appendix* Tables 4 and 5). We follow a similar 20-trial setup in the ablation studies and conduct these

experiments with 5k pre-training iterations, with further details provided in § B.5. Our ablations yield the following insights.

**(i)** Utilizing a binary $G$, the prevalent approach in SSL, leads to a dimensional collapse in our framework, characterized by low feature spreads, diminished ranks, and elevated inter-feature correlations. This stems from the strict enforcement of either identical representations for a pair or none at all. However, we address this by employing finer enforcement in invariance term $\mathcal{L}_{I'}$ through soft $G$ entries. **(ii)** Omitting regularization $\mathcal{L}_R$ leads to the relation graph converging towards a degenerate solution where all $G$ entries collapse to zero. Consequently, no guidance is provided for the invariance term, resulting in dimensional collapse and markedly reduced feature spread. **(iii)** Absence of $\Psi_\psi$ results in dimensional collapse with low spreads. This occurs because the aggregation of various relation graphs remains fixed throughout training, employing $\lambda_i$ in Eq. 7 as hyperparameters. We mitigate this issue by proposing online aggregation through Eq. 8. **(iv)** The SG mechanism prevents $G^{(i)} \in S_G$ and $G$ from collapsing to a degenerate solution, as demonstrated in Cora.

**(v)** Utilizing a standalone $G^k$ results in extremely noisy over-enforcement of neighborhood proximities, leading to representations becoming excessively close, as evidenced by low spread for AmzComp and complete collapse in Cora. **(vi)** Augmentations in $G^a$ prove to be extremely beneficial for Cora and CiteSeer, contradicting the adoption of a completely augmentation-free approach.

**(vii)** In AmzComp, removing $G^a$, $G^{A'}$, $G^L$, or $G^S$ maintains full-rank representations with sufficient spread. Though accuracy slightly drops, we speculate that online clustering compensates for this absence to some extent. The information extracted in $G^O$ can effectively provide some insights comparable to augmentations, adjacency, and PEs. Notably, $G^O$ proves to be the most impactful among other relation graphs in AmzComp, enabling an automatic online approach to capture information that other sources may also offer. This underscores our model's capability to handle various scenarios where higher-order encodings may not be as useful, such as extremely sparse graphs. **(viii)** In AmzComp, the model collapses when we eliminate SE $G^{R'}$, while still incorporating adjacency and PEs. We speculate that RWSE is particularly effective in mitigating noisy information from adjacency and PEs. Consequently, adjacency and PEs should be utilized in conjunction with SEs, leading to our proposal of employing PSEs as the higher-order encodings. Consistent with this rationale, eliminating the higher-order encodings entirely results in generally better performance than removing only one PSE at a time. **(ix)** The noisy nature of $G^R$ is evident in CiteSeer, highlighting the effectiveness of filtering it to construct $G^{R'}$.

**(x)** The profound impact of deep clustering in preventing collapse is evident. Our online clustering maintains a global perspective by evenly distributing representations to learnable prototypes, effectively mitigating potential over-enforcement caused by other relation graphs. **(xi)** Similarly, adopting a global perspective to sparsify $G^O$ yields slightly improved performance. Utilizing the global $F_K$ instead of the local $F_k$ enables the model to capture a broader picture rather than focusing solely on local properties possibly covered by other relation graphs. **(xii)** Enforcing all relation graphs to consider intra-view relations alongside inter-view ones leads to convergence of spread to 1 in fewer iterations and even better downstream performance in Cora, although with a computational overhead.

### 4.4 SSL Pre-Training Analysis

SSL models undergo pre-training for a specific number of iterations before finally being evaluated on a downstream task. However, it is beneficial to examine the pre-training process periodically to gain insights into learning suitable representations. Thus, we evaluated checkpoints at intervals of 1k iterations. Due to the considerably higher resource consumption of CoPhy, pre-training is manually terminated before reaching 7k iterations. Fig. 2a demonstrates the accuracy mean and standard deviation throughout the learning process. The general trend indicates that as the pre-training progresses, the representations tend to become more suitable for the downstream task, prominently observed in AmzComp, AmzPhoto, CoPhy, PubMed, DBLP, CiteSeer, and the initial iterations of CoCS and Cora. Notably, WikiCS exhibits a consistent accuracy for the first few thousand iterations, followed by a decline. In SSL training, the objective may not always align directly with the particular downstream task, as the goal is to derive a more generalized representation adaptable to various downstream tasks. Hence, the decreasing trends after a point suggest that ExGRG may

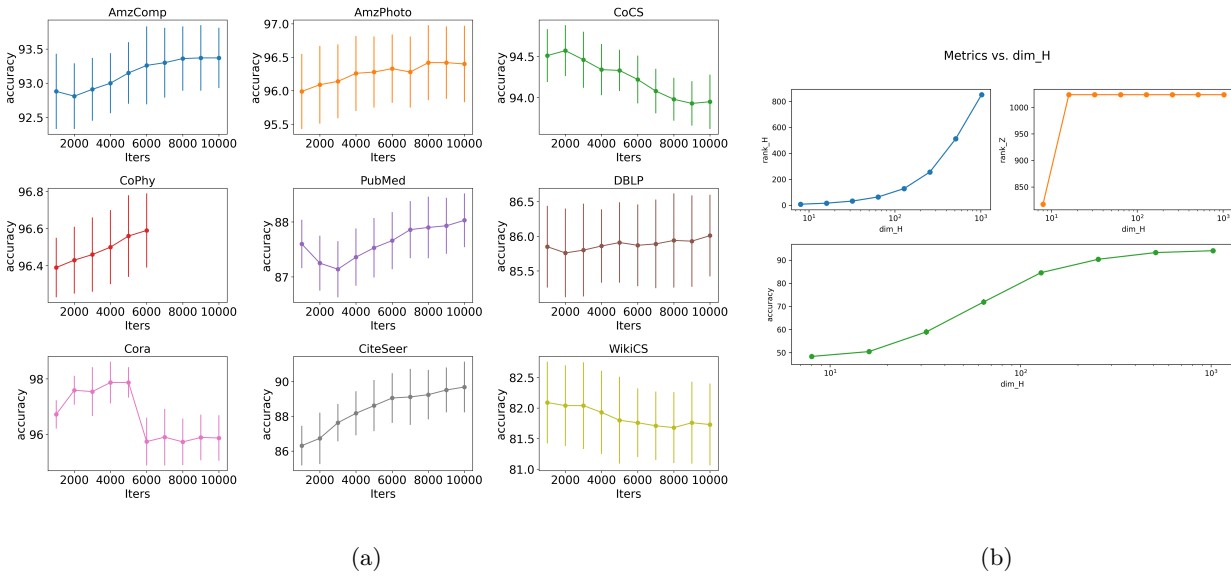

(a) (b)

Figure 2: (a) Accuracy vs. iterations over 20 trials throughout the SSL pre-training. (b) Impact of altering $D_H$ on ranks and downstream accuracy for Amazon Computers.

sacrifice some aspects of a representation suitable for the specific node classification to enhance other aspects targeted in $\mathcal{L}_{ETE}$. Additionally, the overall decreasing trends for correlations, along with the prevailing increasing trends for standard deviations and ranks for both $\boldsymbol{H}$ and $\boldsymbol{Z}$ throughout the pre-training are detailed in *Appendix* Fig.3-8 and § B.6.

### 4.5 Hyperparameter Analysis

The employed hyperparameters, encompassing design choices and optimizers for each dataset, are outlined in *Appendix* Table 6 and § B.7.2. Here, we analyze the influence of these parameters on our model's performance, focusing on AmzComp. Notably, Fig. 2b alters $D_H$, the number of feature dimensions of representations. Both accuracy and rank of $\boldsymbol{H}$ exhibit an upward trend, indicating the model's capacity to effectively leverage the feature dimensions to encode valuable information.

Furthermore, the impact of various parameters is detailed in *Appendix* Fig. 9-13 and § B.7.3. Notably, the simultaneous increase in both $D_Z$ and $D_H$ results in an ascending trend in accuracy and ranks, affirming the model's effectiveness in leveraging larger feature dimensions. Conversely, alternative contrastive methods rapidly saturate, suffering from the curse of dimensionality (Zbontar et al., 2021). In addition, the model exhibits robustness concerning mini-batch size $N$, indicating its ability to derive meaningful directions for guiding the invariance term even with a reduced number of points. Moreover, the influence of $k$ in $\boldsymbol{G}^k$ reveals that extremely high $k$ for kNN negatively impacts performance, contributing to heightened noise in guiding the invariance term. Also, increasing $K$ in $\boldsymbol{G}^O$ emphasizes the stability, but with an overall slightly increasing trend for downstream accuracy. Though we effectively leverage a higher number of prototypes, determining the appropriate number of clusters for each dataset is traditionally challenging (Zhou et al., 2022), underscoring the importance of achieving this general robustness.

## 5 Conclusion

In this paper, we introduced ExGRG, a novel graph SSL approach inspired by the theoretical interplay among the SSL pre-training approaches, the spectral embedding methods in manifold learning, and the EM optimization algorithm. ExGRG offers a comprehensive framework for injecting higher-order domain knowledge into the SSL process. This involves diverse forms of prior knowledge, including positional and

structural encodings and adjacency, alongside online extracted information through a kNN graph and a deep clustering module. To achieve this, we propose explicitly generating a compositional relation graph to guide the invariance objective, deviating from the conventional reliance on data augmentations, which proves problematic and counter-intuitive in the context of the graph domain. Extensive experimental evaluations across diverse graph datasets showcase the superiority of ExGRG over state-of-the-art graph SSL approaches.

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

# A  Method

## A.1  Preliminary

**Nonlinear and Fine-tuning Evaluation Protocols.** Nonlinear probing mirrors the linear probing approach but adopts an MLP or a kNN classifier. In fine-tuning, either all or the last few layers of the pre-trained $f_\theta$ are updated based on backpropagations computed from the downstream task.

**Graph Classification and Link-level Prediction.** Graph downstream tasks encompass node classification, graph classification, or link-level prediction. Graph classification typically involves learning node representations followed by a pooling mechanism to compute the graph representation. Link-level prediction can be simplified into node-level tasks aggregating representations from the two end nodes of a specific link. Hence, while ExGRG primarily focuses on node representation learning, its framework can be extended to facilitate the acquisition of graph-level or node-pairwise representations tailored for link-level tasks.

**LapPE.** This encoding is derived from eigenvectors of the source graph's Laplacian corresponding to the $k$-lowest non-zero eigenvalues (Dwivedi et al., 2023).

**RWSE.** Specifically, the $k$-th RWSE reflects the probability of returning to the starting state of a random walk after precisely $k$ steps (Dwivedi et al., 2021; Liu et al., 2023a).

**SignNet.** Utilizing eigenvectors, SignNet is a sign-invariant network proposed to handle varying numbers of sign-ambiguous eigenvectors (Lim et al., 2022; Rampášek et al., 2022).

## A.2  Explicit Compositional Relation Graph

Our methodology involves initially generating relation graphs from various sources using different construction algorithms. Subsequently, we aggregate all these relation graphs simply through a learnable linear combination to form a comprehensive relation graph $G$, which is then applied in the invariance term. Generally, potential sources for the $G$ generation process encompass representations $Z$, embeddings $H$, node features $X^s$ or $X$, intermediate hidden features, and the embeddings of another latent space projection, such as nodes' PSEs. Some sources, like the adjacency matrix $A^s$ or data augmentation, require no further processing. However, other sources necessitate processing through a so-called construction algorithm, such as kNN or a clustering-based method.

## A.3  Laplacian Eigenmap

$$Z^* = \arg\min_{Z} \operatorname{Tr}\left(Z^T L Z\right) \text{ s.t. } Z^T D Z = I, \tag{11}$$

where $D = VI = 2I$ and $L = D - G$ are degree and Laplcian matrices for $G$, while $Z^*$ is the optimal LE embeddings.

## A.4  Neighborhood Similarity in the Representation Space

Our preliminary experiments indicate that the kNN graph on $H$ provides more informative relations compared to $Z$. We speculate that due to the invariance term, the points in $Z$ are compelled to be excessively close; therefore, the distance between corresponding points in $H$ is more instructive for constructing a relation graph.

## A.5  Higher-Order Graph Encodings

### A.5.1  Adjacency-based Relation Graph

The relation matrix $G^A$, based on the adjacency information of the source graph, is defined in 3.3.2. Intuitively, $G^A$ can provide independent guidance for a standalone relation matrix with increased confidence, especially in datasets exhibiting more homophily. Conversely, using $G^k$ directly in heterophilic graph datasets

might be more beneficial, as all entries in $\boldsymbol{G}^{A'}$ are likely zero with higher confidence. This is because, in a heterophilic graph, neighboring nodes in the input do not share the same class label, making the overlap between nodes that are neighbors in both the representation kNN graph and the source graph less probable.

### A.5.2 PSE-based Relation Graph

The efficacy of various $\boldsymbol{G}^{PSE}$ depends on the dataset and domain under consideration. Our strategy for employing PSEs diverges from their conventional application in transformers. Nevertheless, a similar impact can be observed when utilizing PSEs in transformers, where the performance of RWSE proves advantageous for small molecular data, while LapPE excels in tasks involving image superpixels and those requiring long-range dependencies. In addition, SignNet with DeepSet encodings demonstrates success across diverse domains and tasks (Rampášek et al., 2022; Liu et al., 2023a).

### A.6 Deep Clustering

To obtain the codes $\boldsymbol{Q}$, we follow the approach outlined in Asano et al. (2019) and Caron et al. (2020). This is performed by solving the optimization problem by maximizing the similarity between representations and prototypes, with $\boldsymbol{Q}^*$ representing the optimal solution as

$$\boldsymbol{Q}^* = \arg\max_{\boldsymbol{Q}} \mathrm{Tr}(\boldsymbol{Q}^T \boldsymbol{C} \boldsymbol{H}^T) + \epsilon \mathscr{H}(\boldsymbol{Q}). \tag{12}$$

The term $\mathscr{H}(\boldsymbol{Q}) = -\sum_{i,j} \boldsymbol{Q}_{i,j} \log \boldsymbol{Q}_{i,j}$ represents the entropy of $\boldsymbol{Q}$, while $\epsilon$ serves as a balancing factor, determining the significance of achieving a uniformly distributed partition through entropy maximization. This involves imposing a constraint on the codes $\boldsymbol{Q}$ to lie within the transportation polytope, indicating that our codes $\boldsymbol{Q}$ should adhere to a set $\mathcal{Q}$, where

$$\mathcal{Q} = \left\{ \boldsymbol{Q} \in \mathbb{R}_+^{N \times K} \mid \boldsymbol{Q}^T \mathbf{1}_N = \frac{1}{K} \mathbf{1}_K, \boldsymbol{Q} \mathbf{1}_K = \frac{1}{N} \mathbf{1}_N \right\}. \tag{13}$$

This constraint implies that, on average, each cluster is expected to be assigned approximately $\frac{N}{K}$ data points, reflecting an equal distribution. To determine a suitable $\boldsymbol{Q}$ as a solution for Eq. 12 and 13, we leverage the Sinkhorn-Knopp algorithm (Cuturi, 2013). Therefore, we iteratively calculate two renormalization vectors, $\mathbf{u} \in \mathbb{R}^N$ and $\mathbf{v} \in \mathbb{R}^K$, to address the optimization problem as

$$\boldsymbol{Q}^* = \mathrm{Diag}(\mathbf{u}) \exp\left(\frac{\boldsymbol{C} \boldsymbol{H}^T}{\epsilon}\right) \mathrm{Diag}(\mathbf{v}). \tag{14}$$

### A.7 End-to-End Training Procedure

#### A.7.1 SG Mechanism

In the presence of a gradient path through the invariance term $\mathcal{L}_{I'}$, the optimizer incentivizes the reduction of $\boldsymbol{G}^{(i)}$ entries. However, it is essential to note that these specific entries within $\boldsymbol{G}^{(i)}$ are a refined sparse subset derived from a potentially dense $N \times N$ relation matrix. The existence of such a gradient path to update $\boldsymbol{G}^{(i)}$ generator parameters implies the promotion of weaker relations among instances that initially exhibit strong relations. Inspired by Grill et al. (2020) and Chen & He (2021), to simultaneously optimize both sets of parameters, we propose an SG mechanism specifically on the $\boldsymbol{G}^{(i)}$ generator parameters (Eq. 7), such as the prototypes $\boldsymbol{C}$ for $\boldsymbol{G}^O$, through the $\mathcal{L}_{I'}$ path. Without this SG, we counter-intuitively diminish the $\boldsymbol{G}^{(i)}$ entries for candidate pairs with high potential for closer representations.

#### A.7.2 Relation Matrix Regularization

To ensure the end-to-end aggregation of different relation matrices for deriving the final relation matrix $\boldsymbol{G}$, the aggregator module $\Psi_\psi$ gets updates through the invariance term $\mathcal{L}_{I'}$. The incentive here mirrors the one encountered with each individual relation matrix $\boldsymbol{G}^{(i)}$ in A.7.1, wherein, lacking constraints, the optimizer tends to converge all $\boldsymbol{G}$ entries to a trivial solution. To mitigate this, we propose the incorporation of a

Table 3: Statistics of the employed datasets.

| Statistic | WikiCS | AmzComp | AmzPhoto | CoCS | CoPhy | Cora | CiteSeer | PubMed | DBLP |
|---|---|---|---|---|---|---|---|---|---|
| # Nodes | 11,701 | 13,752 | 7,650 | 18,333 | 34,493 | 2,708 | 3,327 | 19,717 | 17,716 |
| # Edges | 216,123 | 574,418 | 287,326 | 327,576 | 991,848 | 5,429 | 4,732 | 44,338 | 105,734 |
| # Classes | 10 | 10 | 8 | 15 | 5 | 7 | 6 | 3 | 4 |
| # Features | 300 | 767 | 745 | 6,805 | 8,451 | 1,433 | 3,703 | 500 | 1,639 |

regularizer aimed at discouraging the entries of $\boldsymbol{G}$ from collapsing to the degenerate solution, wherein all entries converge to zero. This regularizer is formulated based on the concept of the summation of all $\boldsymbol{G}$ entries demonstrated in Eq. 9.

### A.7.3 Overall Multi-term Loss Objective

During optimization of $\mathcal{L}_{ETE}$, parameters $\theta$ of the encoder $f_\theta$ and $\phi$ of the expander $g_\phi$ undergo updates through $\mathcal{L}_V$, $\mathcal{L}_C$, $\mathcal{L}_{I'}$, and $\mathcal{L}_O$. The prototypes $\boldsymbol{C}$ receive updates through $\mathcal{L}_O$, while the aggregation module parameters, including $\psi$ for $\Psi_\psi$, are updated through $\mathcal{L}_R$ and invariance term $\mathcal{L}_I$.

## B Experimental Evaluation

### B.1 Datasets

Our node classification experiments encompass evaluations on 9 datasets: WikiCS, Amazon Computers (AmzComp), Amazon Photo (AmzPhoto), Coauthor CS (CoCS), Coauthor Physics (CoPhy), Cora, Citeseer, PubMed, and DBLP. The corresponding statistics for these datasets, such as the number of nodes, edges, classes, and features, are provided in Table 3. Furthermore, if a specific dataset is not used to evaluate a particular method, the corresponding cell is left blank in Table 1.

### B.2 SSL Encoder

We employ the GCN proposed in Kipf & Welling (2016a) as the SSL encoder $f_\theta$. Specifically, the output of the $l$-th GCN layer, $H^{(l)}$, is expressed as

$$H^{(l)} = \text{GCN}^{(l)}(\boldsymbol{X}, \boldsymbol{A}) = \sigma(\hat{\boldsymbol{D}}^{-1/2}\hat{\boldsymbol{A}}\hat{\boldsymbol{D}}^{-1/2}\boldsymbol{X}\boldsymbol{W}^{(l)}). \tag{15}$$

Here, $\hat{\boldsymbol{A}} = \boldsymbol{A} + \boldsymbol{I}$ represents the adjacency matrix in the presence of self-loops. Additionally, we denote the degree matrix as $\hat{\boldsymbol{D}} = \sum_i \hat{\boldsymbol{A}}_i$, nonlinear activation function as $\sigma$, and trainable parameters as $\boldsymbol{W}^{(l)}$, eventually forming parameters $\theta$.

### B.3 Downstream Evaluation

To employ an SSL model for a downstream node classification task, the encoder $f_\theta$ and expander $g_\phi$ undergo joint training without utilizing labels $\boldsymbol{Y}^s$, effectively following an unsupervised approach for a predetermined number of iterations. Subsequently, the expander $g_\phi$ is discarded, likely to leverage more generalized representations. The fixed representations obtained from the encoder $f_\theta$, in addition to the labels $\boldsymbol{Y}^s$, are employed to train a logistic regression classifier in the graph domain in contrast to the linear layer counterpart used in vision applications (Thakoor et al., 2021). This classifier is trained on top of the frozen representations and, subsequently, evaluated on a validation split to determine optimal hyperparameters. Finally, the node classification accuracy is assessed on a separate test set. This entire process is repeated for 20 trials, each characterized by random model initialization and different train-validation-test splits. The randomness is introduced due to the notable variability observed in computed accuracy across different splits in graphs.

Table 4: Ablation studies on CiteSeer. Models are altered concerning ExGRG (5k iters) as the reference.

| [Ablated] Model | Accuracy | corr $H$ | corr $Z$ | std $H$ | std $Z$ | nstd $H$ | rank $H$ | rank $Z$ |
|---|---|---|---|---|---|---|---|---|
| ExGRG (10k iters) | 89.68 ± 1.46 | 0.0000 | 0.0163 | 0.0547 | 0.3335 | 0.0233 | 956 | 976 |
| ExGRG (5k iters) | 88.61 ± 1.47 | **0.0001** | 0.0138 | 0.0514 | **0.2799** | **0.0211** | **894** | **976** |
| Binary $G$ | 84.57 ± 1.11 | **0.1491** | 0.0004 | 0.1150 | **0.0509** | 0.0258 | **237** | **627** |
| No $\mathcal{L}_R$ | 83.96 ± 1.51 | **1.1409** | 0.0000 | 0.1847 | **0.0128** | 0.0231 | **103** | **578** |
| No $\Psi_\psi$ | 84.05 ± 1.43 | **1.1408** | 0.0000 | 0.1847 | **0.0127** | 0.0231 | **103** | **588** |
| No $F_{SG}$ on $G^S$ | 88.48 ± 1.42 | 0.0001 | 0.0135 | 0.0512 | 0.2759 | 0.0210 | 890 | 972 |
| No Standalone $G^k$ | 88.47 ± 1.42 | 0.0001 | 0.0135 | 0.0512 | 0.2759 | 0.0210 | 890 | 972 |
| No $G^{A'}$ or $G^{R'}$ | 86.69 ± 1.12 | 0.0002 | 0.0030 | 0.0526 | **0.1747** | **0.0193** | 791 | 988 |
| No $G^a$ | 84.35 ± 1.32 | **1.2491** | 0.0000 | 0.1852 | **0.0126** | 0.0232 | **99** | **598** |
| No $G^{A'}$ | 84.20 ± 1.27 | **1.2497** | 0.0000 | 0.1853 | **0.0125** | 0.0232 | **99** | **618** |
| No $G^L$ | 85.67 ± 1.19 | 0.0002 | 0.0054 | 0.0540 | **0.1899** | 0.0208 | 737 | 902 |
| No $G^{R'}$ | 87.92 ± 1.51 | **0.0001** | 0.0124 | 0.0560 | 0.2697 | 0.0216 | 864 | **969** |
| $G^R$ instead of $G^{R'}$ | 83.98 ± 1.79 | **0.5892** | 0.0000 | 0.1579 | **0.0146** | **0.0279** | **110** | 1043 |
| No $G^S$ | 88.11 ± 1.44 | 0.0001 | 0.0125 | 0.0568 | 0.2702 | 0.0214 | 862 | 971 |
| No $G^{PSE}$ or $G^{A'}$ | 87.90 ± 1.48 | 0.0001 | 0.0124 | 0.0560 | 0.2697 | 0.0215 | 864 | 969 |
| No $G^O$ | 88.57 ± 1.47 | 0.0001 | 0.0120 | 0.0521 | **0.2671** | 0.0204 | 883 | 974 |
| $F_k$ instead of $F_K$ in $G^O$ | 88.48 ± 1.42 | 0.0001 | 0.0135 | 0.0512 | 0.2759 | 0.0210 | 890 | 972 |
| $S_O = \{(i,j)|\ i = j\}$ | 88.50 ± 1.42 | 0.0001 | 0.0135 | 0.0512 | 0.2759 | 0.0210 | 890 | 972 |
| $S_O = S_a$ | 88.35 ± 1.25 | 0.0001 | 0.0135 | 0.0512 | 0.2759 | 0.0210 | 890 | 972 |
| Intra Relations | 89.38 ± 1.25 | 0.0000 | 0.0159 | 0.0542 | **0.3273** | 0.0232 | 955 | 975 |

## B.4 Additional Evaluation Metrics

In certain instances, the accuracy of the graph datasets exhibits a degree of resilience to alterations in the model. This observation underscores the need to consider our defined metrics for a more comprehensive comparison of different scenarios, including our ablated models. These metrics are computed on both $H$ and $Z$, providing deeper insights into their characteristics. Reported evaluation metrics are inspired by how VICReg criteria in Eq. 1 contribute to constructing the representations.

The **corr** metric measures a notion of correlation among different pairs of features. This is inspired by $\mathcal{L}_C$, representing the average over squared off-diagonal entries of the covariance matrix corresponding to each pair of feature dimensions. The **std** metric calculates the average standard deviation along each feature dimension. The **nstd** metric captures the same notion of average standard deviation as *std* but on the $L_2$-normalized representations. Given the utilization of Euclidean similarity in constructing the embeddings $Z$, vector norms also encode information. Therefore, **nstd** is exclusively reported for $H$. The **rank** metric corresponds to the rank of the representations, capturing dimensional collapse. Generally, a desirable model exhibits higher values for accuracy's mean, std, nstd, and rank while demonstrating lower values for corr and accuracy's standard deviation.

## B.5 Ablation Studies

Followings are the direct impact of our ablations on the overall performance, including downstream accuracy and our defined metrics in § B.4. These are reported based on the ablations on Amazon Computers in Table 2 and slightly differ from observations in CiteSeer (Table 4) and Cora (Table 5).

**ExGRG ($X$k iters)** signifies the reference model trained with $X$ thousand(s) iterations. To maintain consistency in comparison, all ablated models listed in other rows should be assessed relative to *ExGRG (5k iters)*, ensuring an equal number of iterations for reference. The **Binary $G$** model enforces strict binary values (0 or 1) for all entries in $G$, unlike the soft version of ExGRG. This results in a slight decrease in accuracy,

Table 5: Ablation studies on Cora. Models are altered concerning ExGRG (5k iters) as the reference.

| [Ablated] Model | Accuracy | corr $H$ | corr $Z$ | std $H$ | std $Z$ | nstd $H$ | rank $H$ | rank $Z$ |
|---|---|---|---|---|---|---|---|---|
| **ExGRG (5k iters)** | $97.87 \pm 0.55$ | **0.0000** | 0.0935 | 0.0281 | **0.6168** | 0.0320 | 946 | 824 |
| **Binary $G$** | $93.84 \pm 0.99$ | **5.4081** | 0.0068 | 0.2642 | **0.0661** | 0.0121 | **47** | **43** |
| **No $\mathcal{L}_R$** | $93.57 \pm 1.09$ | **73.106** | 0.0065 | 0.4721 | **0.0604** | 0.0272 | **9** | **20** |
| **No $\Psi_\psi$** | $93.61 \pm 1.15$ | **70.161** | 0.0067 | 0.4635 | **0.0607** | 0.0275 | **9** | **21** |
| **No $F_{SG}$ on $G^S$** | $93.60 \pm 1.25$ | 0.0391 | 0.0061 | 0.0897 | **0.0662** | 0.0211 | **217** | **341** |
| **Add Standalone $G^k$** | $93.17 \pm 1.16$ | **11.229** | 0.0131 | 0.2841 | **0.1083** | 0.0171 | **23** | **61** |
| **No $G^{A'}$ or $G^{R'}$** | $93.15 \pm 1.19$ | **127.182** | 0.0072 | 0.5422 | **0.0609** | 0.0229 | **8** | **21** |
| **No $G^a$** | $93.01 \pm 1.17$ | **14.856** | 0.0081 | 0.3121 | **0.0639** | 0.0258 | **13** | **19** |
| **No $G^{A'}$** | $93.18 \pm 1.26$ | **11.759** | 0.0086 | 0.2942 | **0.0649** | 0.0260 | **15** | **20** |
| **No $G^L$** | $95.65 \pm 1.13$ | **0.2911** | 0.1888 | 0.1281 | **0.2483** | 0.0118 | **230** | **435** |
| **No $G^{R'}$** | $96.16 \pm 1.03$ | 0.0059 | 0.0646 | 0.0633 | **0.3737** | 0.0178 | **571** | 784 |
| **$G^R$ instead of $G^{R'}$** | $93.20 \pm 1.35$ | 0.0386 | 0.0069 | 0.0914 | **0.0679** | 0.0248 | **215** | **305** |
| **No $G^S$** | $92.79 \pm 1.13$ | 0.0462 | 0.0086 | 0.0933 | **0.0728** | 0.0249 | **154** | **117** |
| **No $G^{PSE}$ or $G^{A'}$** | $95.91 \pm 0.97$ | 0.0067 | 0.0398 | 0.0662 | **0.2916** | 0.0172 | **570** | 783 |
| **No $G^O$** | $93.10 \pm 0.81$ | **0.3040** | 0.0082 | 0.1174 | **0.0666** | 0.0259 | **91** | **108** |
| **$F_k$ instead of $F_K$ in $G^O$** | $93.43 \pm 1.06$ | 0.0356 | 0.0203 | 0.1039 | **0.1618** | 0.0147 | **272** | **439** |
| **$S_O = \{(i,j)|\ i = j\}$** | $93.20 \pm 1.24$ | 0.0660 | 0.0062 | 0.0919 | **0.0639** | 0.0259 | **176** | **280** |
| **$S_O = S_a$** | $96.17 \pm 0.91$ | **0.1995** | 0.0217 | 0.1089 | **0.1142** | 0.0117 | **256** | **462** |
| **Intra Relations** | $95.24 \pm 1.07$ | 0.0364 | 0.0219 | 0.0712 | **0.1030** | 0.0155 | **220** | **236** |

coupled with a notable increase in corr $H$, significantly lower std $Z$, lower ranks, and slightly decreased nstd. **No $\mathcal{L}_R$** eliminates the $G$ regularization loss, leading to reduced accuracy, notably increased corr $H$, a substantial decrease in std $Z$, a slight dip in nstd, and significantly lower ranks, indicating a dimensional collapse. **No $\Psi_\psi$** removes the online aggregator $\Psi_\psi$, implying that parameters $\lambda_i$ become hyperparameters, fixed throughout training. This results in diminished accuracy, elevated corr $H$, collapse indicated by low ranks, and reduced std $Z$ and nstd. **No $F_{SG}$ on $G^S$** signifies the absence of the SG mechanism on SignNet, allowing it to be altered from its pre-trained status. This leads to a marginal decrease in accuracy, while the other metric values remain largely unchanged.

**Add/No Standalone $G^k$** signifies the addition or removal of a standalone relation matrix $G^k$ solely based on kNN in $H$. The inclusion of $G^k$ depends on the extent to which the unfiltered kNN graph proves beneficial for a particular dataset. For AmzComp, incorporating $G^k$ leads to reduced accuracy and a slight decrease in std $Z$. **No $G^{A'}$ or $G^{R'}$** removes any relation graph derived from $G^k$, encompassing the exclusion of $G^A$ or, in some datasets, the absence of $G^R$, with details in *Appendix* Table 6 and § B.7.2. This leads to a minor decrease in accuracy and a slight reduction in std $Z$. **No $G^a$** eliminates the relation graph directly based on pairwise augmentation. In this scenario, where the enforcement of identical representations for pairs in $S_a$ is not explicitly present, random augmentations are still applied to two views. Thus, other components, such as $G^k$ may indirectly enforce similarity for pairs in $S_a$. This leads to a slight decrease in accuracy and a minor reduction in std $Z$.

**No $G^{A'}$** excludes the utilization of adjacency information for constructing a relation matrix. This leads to a slight decrease in accuracy and std $Z$. **No $G^L$** eliminates the relation graph based on Laplacian encoding, leading to a marginal decrease in accuracy and std $Z$. **No $G^{R'}$** eliminates the relation graph based on random walk encoding, resulting in significantly higher corr $H$, reduced accuracy, std $Z$, nstd $H$, and a collapse in ranks. **$G^R$ instead of $G^R$** utilizes the unfiltered version of $G^R$ as opposed to filtering it by computing $G^R \odot G^k$ with details in *Appendix* Table 6 and § B.7.2. This leads to decreased accuracy and reduced std $Z$. **No $G^S$** eliminates the relation graph based on SignNet encoding, leading to a slight decrease in accuracy and a minor reduction in std $Z$. **No $G^{PSE}$ or $G^{A'}$** omits any utilization of positional or

structural properties of the graph, involving neither PSE nor adjacency. This leads to a minor decrease in accuracy and a slight reduction in std $\boldsymbol{Z}$.

**No $\boldsymbol{G}^O$** entirely removes the end-to-end clustering module. This leads to decreased accuracy, notably increased corr $\boldsymbol{H}$, and significantly reduced std $\boldsymbol{Z}$. $F_k$ **instead of** $F_K$ **in $\boldsymbol{G}^O$** employs an alternative method to sparsify $\boldsymbol{G}^O$ by utilizing the local version $F_k$ instead of the global $F_K$ in Eq. 6. In our proposed setting, the top-K globally highest entries in the entire relation graph are selected, and the rest are set to zero. Here, a kNN-like operation is performed, where, for each data point as an anchor, the top-K largest entries are selected, and the rest are set to zero. This leads to a slight decrease in accuracy and std $\boldsymbol{Z}$. $S_O = \{(i,j)|\ i = j\}$ and $S_O = S_a$ yield similar outcomes, respectively eliminating augmentation pairs $S_a$ and self pairs $(i = j)$ from $S_O$. The set $S_O$ denotes the pairs for applying distribution alignment in the optimal transport loss function. This includes both *self* data points, i.e., $i = j$, and *augmentation* pairs $(i, j)$. These ablations each eliminate one of these components in the model. This leads to a slight decrease in accuracy and std $\boldsymbol{Z}$.

### B.5.1   Intra-view Relations

In an augmentation-based relation graph, only a subset of inter-view entries is non-zero, and all intra-view entries are zero. However, our proposed approach for constructing relation graphs extends this by incorporating both inter-view and intra-view relations, thereby capturing a broader range of possible interactions.

The compositional relation graph $\boldsymbol{G}$ consistently integrates intra-view relations. However, for the Neighborhood Similarity graph $\boldsymbol{G}^k$, our empirical analysis indicates that considering only inter-view relations is sufficient to maintain comparable performance without the computational overhead of including intra-view relations. In this context, the remaining graphs $\boldsymbol{G}^{(i)} \in S_G$, alongside the compositional graph $\boldsymbol{G}$, still contain non-zero intra-view relations. We hypothesize that inter-view relations alone suffice in the Neighborhood Similarity graph because the k-nearest neighbors identified from the alternate view already provide sufficiently strong neighboring relations. Consequently, intra-view neighbors are not necessary to enforce the invariance term effectively.

**Intra Relations** in our ablation studies represents a model where $\boldsymbol{G}^k$ is forced to consider both inter- and intra-view relations. With a fixed GPU memory allocation, the impact of this proposed approach becomes more pronounced in smaller datasets such as CiteSeer and Cora in *Appendix* Tables 4 and 5. The introduction of intra-view relations has the potential to result in the construction of better representations, manifesting in higher accuracy and increased std $\boldsymbol{Z}$.

### B.6   Pre-Training Analysis

This section serves as an extension to § 4.4, delving into the trends observed in correlations, standard deviations, and ranks for representations $\boldsymbol{H}$ and embeddings $\boldsymbol{Z}$.

**Correlation.** Fig. 3 and 4 illustrate our defined corr metric, resembling the correlation among various pairs of feature dimensions for $\boldsymbol{H}$ and $\boldsymbol{Z}$. Although $\mathcal{L}_C$ targets reducing correlation for $\boldsymbol{Z}$, we observe that the correlation among feature dimensions of $\boldsymbol{H}$ also decreases over iterations. The metric corr $\boldsymbol{Z}$ initially experiences a significant decrease and then slightly increases throughout the training. We speculate this behavior is because the correlation is slightly sacrificed after a few thousand iterations to address other aspects of a proper representation.

**Standard Deviation.** In Fig. 5 and 6, our *nstd* and *std* metrics for $\boldsymbol{H}$ and $\boldsymbol{Z}$ are presented. A higher standard deviation for representations indicates a wider spread of data points in each feature dimension, reflecting a notion of points' separability along a particular dimension. The notable drop in std $\boldsymbol{Z}$ for Cora aligns with the decrease in accuracy, suggesting that this metric could be an indicator to terminate SSL pre-training. Our ultimate target for std $\boldsymbol{Z}$ is 1, corresponding to $\mathcal{L}_V$. Fig. 6 demonstrates that certain datasets achieve this target within 10k iterations while others progressively move toward this objective throughout the pre-training. Additionally, despite $\mathcal{L}_V$ loss being applied to $\boldsymbol{Z}$, Fig. 5 reveals a more or less increasing trend for nstd $\boldsymbol{H}$.

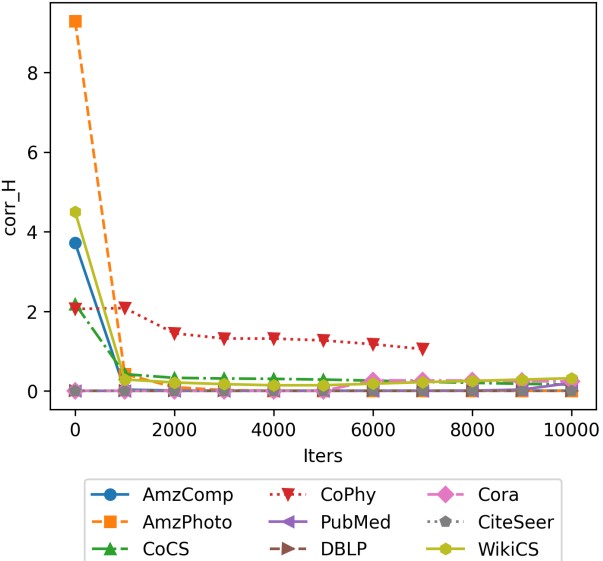

Figure 3: *corr $\boldsymbol{H}$* vs. iterations throughout the SSL pre-training.

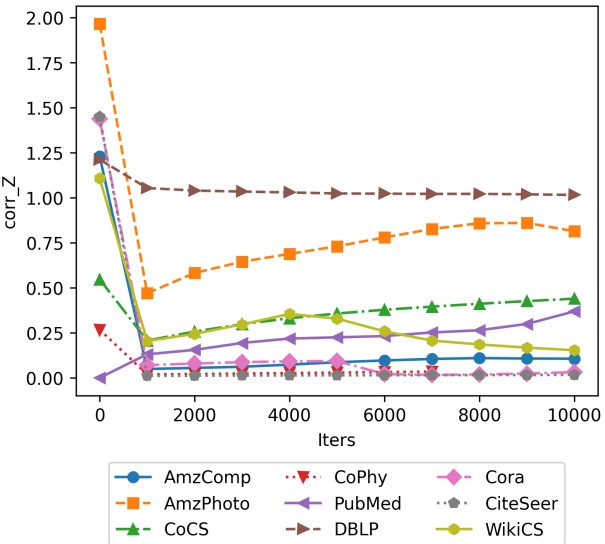

Figure 4: *corr $\boldsymbol{Z}$* vs. iterations throughout the SSL pre-training.

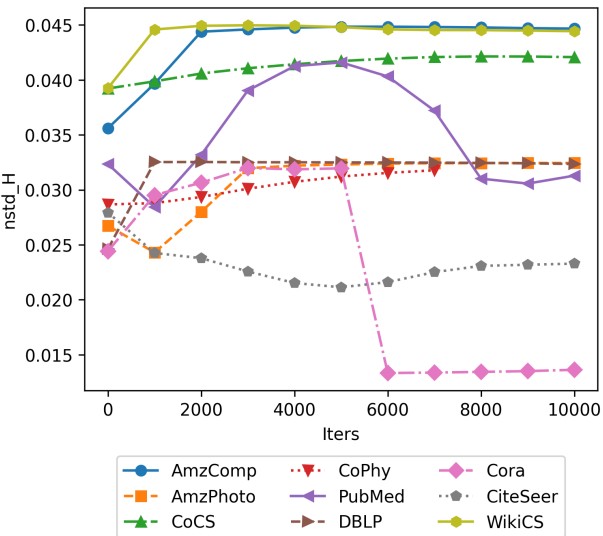

Figure 5: *nstd* $\boldsymbol{H}$ vs. iterations throughout the SSL pre-training.

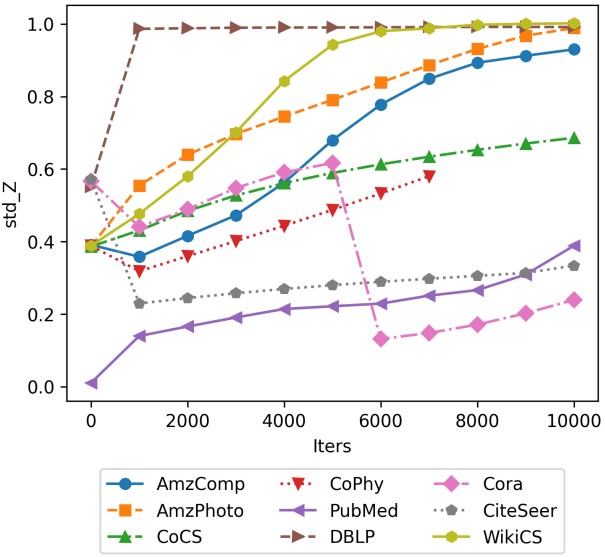

Figure 6: *std* $\boldsymbol{Z}$ vs. iterations throughout the SSL pre-training.

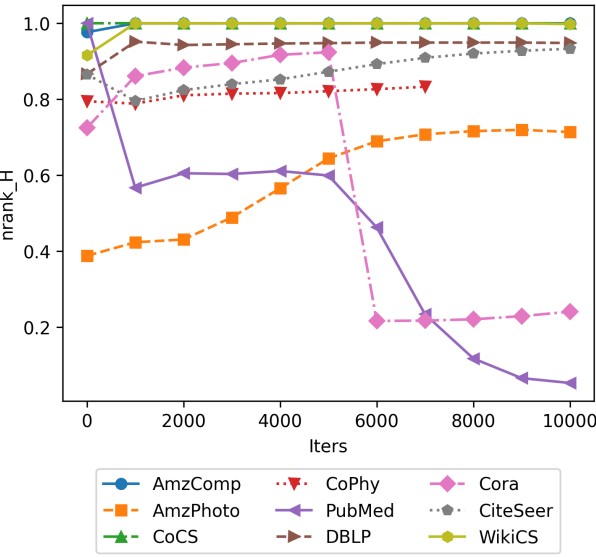

Figure 7: *nrank* $\boldsymbol{H}$ vs. iterations throughout the SSL pre-training.

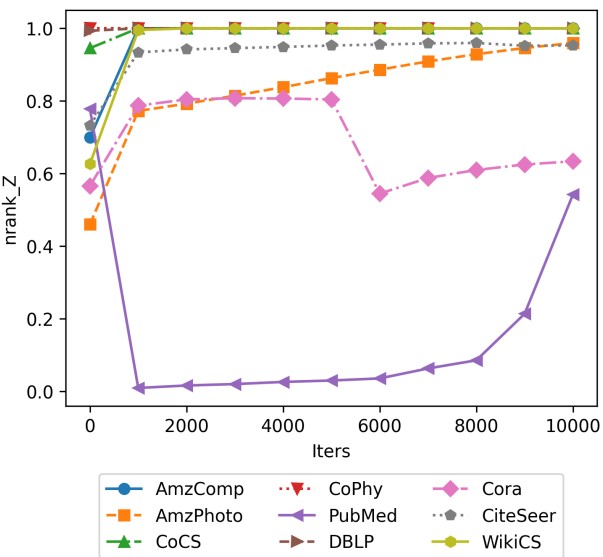

Figure 8: *nrank* $\boldsymbol{Z}$ vs. iterations throughout the SSL pre-training.

**Rank.** Fig. 7 and 8 demonstrates a notion of rank for $\boldsymbol{H}$ and $\boldsymbol{Z}$. This is essentially the normalized version of the introduced *rank* discussed in § B.4, obtained by dividing the rank by the corresponding $D_H$ or $D_Z$. Higher ranks signify a more effective utilization of the capacity of the feature dimensions to map the points. With the exception of PubMed, Fig. 8 demonstrates a consistent increase in *nrank*, showcasing the effectiveness of $L_{ETE}$. We speculate that the PubMed trend suggests that necessary information could be encoded into lower dimensions, a phenomenon also reflected in nrank $\boldsymbol{H}$ from Fig. 7. For PubMed, nrank $\boldsymbol{Z}$ increases after an initial drop, closely mirroring the trend observed in accuracy. The corresponding drops in Cora are also likely attributable to over-training, as evidenced by the corresponding accuracy trend on this dataset.

### B.7 Hyperparameters

### B.7.1 Hyperparameter Tuning

Hyperparameter tuning is conducted to identify optimal parameters for each utilized dataset. We adopt a procedure consistent with prior graph SSL methods, notably BGRL (Thakoor et al., 2021). For each dataset, hyperparameters are selected using a single train/validation/test partition. We employ the Bayesian Optimization algorithm implemented in Ray Tune (Liaw et al., 2018) to explore the hyperparameter space efficiently. As is standard practice, the validation split is utilized to select hyperparameters that maximize performance on the downstream task for that split. The performance metrics reported in Table 1 are based on evaluations conducted on the fixed test split, averaged across 20 random model initializations, and varying selections of the training and validation splits.

In the context of SSL, representation learning and evaluation are typically divided into two phases: pre-training and training. During the pre-training phase, the encoder and expander weights are learned in a self-supervised manner, i.e., without the use of labels. In the subsequent training phase, the expander is replaced with a classifier, which is then trained on the downstream task while the encoder's weights are kept fixed. SSL algorithms, including ExGRG, emphasize the pre-training phase. Hyperparameters are divided into two categories: those relevant to the pre-training phase and those associated with training, such as the hyperparameters for the L2-regularized logistic regression classifier used in downstream tasks.

To ensure a fair comparison with baseline methods, our experimental methodology adheres to standard practices in the graph SSL literature. To determine the optimal pre-training hyperparameters, we use a fixed random partitioning of the dataset into train/validation/test splits and tune the pre-training hyperparameters based on the performance of the validation split in the downstream task. After pre-training, a classifier is trained to assess the downstream performance for each candidate hyperparameter configuration. The training hyperparameters of the classifier are then optimized to achieve the highest validation accuracy with the fixed pre-trained encoder weights. This process is repeated iteratively: pre-training hyperparameters are selected, pre-training is performed, and then the classifier is trained by optimizing its training hyperparameters on the validation split. Validation performance for each candidate set of pre-training hyperparameters is reported before moving to the next set.

Subsequently, the pre-training hyperparameters remain fixed, as determined previously. Afterwards, the pre-training step is performed once, after which the encoder's weights are frozen. We then conduct the downstream training step over 20 trials with different model initializations and distinct train/validation splits. This approach accounts for performance variability due to different partitioning in node classification datasets. In each trial, we optimize the training hyperparameters on the trial-specific validation split and subsequently report the performance on the fixed test set.

### B.7.2 Selected Hyperparameters

The chosen pre-training hyperparameters are outlined in Table 6, corresponding to the models whose performance is detailed in Table 1. Due to constraints on the GPU with a 12 GB memory, a limited number of nodes is randomly selected at each iteration to construct the relation graph $\boldsymbol{G}$; this quantity is denoted as the mini-batch size $N$ ($\boldsymbol{G}$ *B.S.*). The optimizer algorithm utilized for end-to-end model training, including AdamW (Loshchilov & Hutter, 2017) or Adam (Kingma & Ba, 2014) algorithms, is denoted as *Optim.* The learning rate is represented by *L.R.*, and *#Iters.* signifies the number of iterations used for training the SSL

Table 6: Selected hyperparameters utilized for each dataset.

| HP | WikiCS | AmzComp | AmzPhoto | CoCS | CoPhy | Cora | CiteSeer | PubMed | DBLP |
|---|---|---|---|---|---|---|---|---|---|
| $N$ ($G$ **B.S.**) | 3072 | 3072 | 3072 | 2560 | 2048 | 3072 | 2048 | 2560 | 3072 |
| **Optim.** | AdamW | AdamW | AdamW | AdamW | AdamW | Adam | Adam | Adam | Adam |
| **L.R.** | 1e-4 | 1e-4 | 5e-5 | 5e-6 | 1e-5 | 5e-4 | 5e-5 | 1e-4 | 5e-4 |
| **#Iters.** | 1000 | 9000 | 9000 | 2000 | 6000 | 5000 | 10000 | 10000 | 10000 |
| $\alpha$ $[\mathcal{L}_V]$ | 40 | 100 | 80 | 60 | 60 | 100 | 40 | 80 | 100 |
| $\beta$ $[\mathcal{L}_C]$ | 10 | 80 | 10 | 10 | 100 | 80 | 100 | 10 | 10 |
| $\gamma$ $[\mathcal{L}_{I'}]$ | 5 | 5 | 5 | 5 | 5 | 5 | 5 | 5 | 5 |
| $[f_\theta]$ **Act.** | PReLU | PReLU | PReLU | PReLU | PReLU | ReLU | PReLU | ReLU | ReLU |
| $[f_\theta]$ **Norm.** | BN | BN | BN | BN | BN | BN | None | BN | None |
| $[f_\theta]$ **#Lay.** | 2 | 2 | 2 | 2 | 2 | 2 | 2 | 2 | 2 |
| $[f_\theta]$ **#Hid.** | 1024 | 1024 | 1024 | 1024 | 1024 | 1024 | 1024 | 1024 | 1024 |
| $D_H$ | 512 | 512 | 1024 | 512 | 512 | 1024 | 1024 | 512 | 1024 |
| $[g_\phi]$ **Act.** | PReLU | PReLU | PReLU | PReLU | PReLU | ELU | ELU | ELU | ELU |
| $[g_\phi]$ **Norm.** | BN | BN | BN | BN | None | BN | BN | None | BN |
| $[g_\phi]$ **#Lay.** | 2 | 2 | 2 | 2 | 2 | 2 | 2 | 2 | 2 |
| $[g_\phi]$ **#Hid.** | 2048 | 2048 | 1024 | 2048 | 512 | 1024 | 2048 | 512 | 512 |
| $D_Z$ | 1024 | 1024 | 1024 | 1024 | 128 | 1024 | 2048 | 1024 | 1024 |
| $\alpha_2$ $[\mathcal{L}_R]$ | 0.05 | 0.02 | 0.5 | 0.2 | 2 | 0.5 | 0.02 | 0.2 | 2 |
| $[\Psi_\psi]$ **#Lay.** | 3 | 3 | 3 | 3 | 3 | 4 | 2 | 3 | 4 |
| $[\Psi_\psi]$ **#Hid./#In** | 2 | 2 | 2 | 2 | 2 | 2 | 2 | 2 | 2 |
| $[\boldsymbol{G}^a]$ $p_1^e$ | 0.2 | 0.5 | 0.4 | 0.3 | 0.4 | 0.2 | 0.2 | 0.4 | 0.1 |
| $[\boldsymbol{G}^a]$ $p_1^n$ | 0.2 | 0.2 | 0.1 | 0.3 | 0.1 | 0.3 | 0.3 | 0 | 0.1 |
| $[\boldsymbol{G}^a]$ $p_2^e$ | 0.3 | 0.4 | 0.1 | 0.2 | 0.1 | 0.4 | 0 | 0.1 | 0.4 |
| $[\boldsymbol{G}^a]$ $p_2^n$ | 0.1 | 0.1 | 0.2 | 0.4 | 0.4 | 0.4 | 0.2 | 0.2 | 0 |
| **Standalone $\boldsymbol{G}^k$** | False | False | False | False | True | False | True | False | False |
| $[\boldsymbol{G}^k]$ **k** | 10 | 12 | 80 | 8 | 64 | 32 | 8 | 4 | 32 |
| $[\boldsymbol{G}^L]$ **k** | 32 | 16 | 8 | 48 | 4 | 8 | 24 | 8 | 48 |
| $[\boldsymbol{G}^L]$ **Freq.** | 40 | 56 | 8 | 32 | 16 | 32 | 64 | 48 | 40 |
| $[\boldsymbol{G}^R]$ **k** | 40 | 64 | 16 | 8 | 32 | 80 | 40 | 56 | 56 |
| $[\boldsymbol{G}^R]$ **Kernel** | 20 | 16 | 8 | 16 | 8 | 24 | 16 | 12 | 20 |
| $\boldsymbol{G}^{R'}$ **instead of $\boldsymbol{G}^R$** | True | True | True | True | False | False | True | True | True |
| $[\boldsymbol{G}^S]$ **k** | 48 | 32 | 80 | 4 | 8 | 4 | 8 | 32 | 16 |
| $[\boldsymbol{G}^S]$ **Freq.** | 8 | 16 | 24 | 10 | 20 | 10 | 8 | 12 | 10 |
| $[\boldsymbol{G}^S]$ **Arch.** | DeepSet | DeepSet | MLP | MLP | MLP | DeepSet | MLP | MLP | MLP |
| $[\boldsymbol{G}^O]$ **K** | 64 | 64 | 64 | 64 | 64 | 64 | 128 | 64 | 64 |
| $[\boldsymbol{G}^O]$ $\tau$ | 0.1 | 0.1 | 0.1 | 0.1 | 0.1 | 0.1 | 0.1 | 0.1 | 0.1 |
| $[\boldsymbol{G}^O]$ **S.K. #Iters.** | 6 | 6 | 6 | 6 | 6 | 6 | 6 | 6 | 6 |
| $[\boldsymbol{G}^O]$ $\epsilon$ | 0.05 | 0.05 | 0.05 | 0.05 | 0.05 | 0.05 | 0.05 | 0.05 | 0.05 |
| $[\boldsymbol{G}^O]$ $K_g/N$ | 32 | 8 | 4 | 14 | 48 | 12 | 32 | 8 | 32 |
| $\alpha_1[\mathcal{L}_O]$ | 1 | 0.01 | 0.2 | 0.01 | 2 | 0.2 | 0.01 | 0.02 | 0.5 |

encoder and expander without utilizing labels. No learning scheduler is employed. In the downstream evaluation, mirroring methods like BGRL (Thakoor et al., 2021), the expander is replaced with an $L_2$-regularized logistic regression classifier, while the encoder weights remain fixed.

The coefficients for the variance, covariance, and invariance loss terms introduced in Eq. 1 are denoted by $\alpha$, $\beta$, and $\gamma$, respectively. We employed various activation functions in the SSL encoder and expander, including PReLU (He et al., 2015), ReLU, and ELU (Clevert et al., 2015). Additionally, for normalization techniques, we utilized BN (Batch Normalization) (Ioffe & Szegedy, 2015) or None (no normalization). The parameters $[f_\theta]$ *Act.*, *Norm.*, *#Lay.*, *#Hid.*, and $D_H$ denote the SSL encoder's activation function, normalization technique, number of GCN layers, number of features in hidden layers, and representation dimension, i.e., the last GCN output dimension, respectively. The same set of hyperparameters is reported for the expander $g_\phi$ modeled with an MLP.

The parameter $\alpha_2$ serves as the coefficient for the relation graph regularization loss term $\mathcal{L}_R$. Parameters $[\Psi_\psi]$ *#Lay.* and *#Hid./#In* represent the number of layers and the proportion of the hidden to the input dimension of the relation graph aggregator $\Psi_\psi$.

The augmentation hyperparameters, starting with $\boldsymbol{G}^a$, are directly inherited from BGRL (Thakoor et al., 2021). The parameters $p_i^e$ and $p_i^n$ represent the probabilities associated with masking edges and node features in the $i$-th SSL view.

*Standalone* $\boldsymbol{G}^k$ signifies whether a relation graph exclusively derived from a kNN on representations, denoted as $\boldsymbol{G}^k$, is directly utilized. Depending on the noise level in this guidance, its inclusion may enhance performance in certain datasets. For datasets where this parameter is set as *False*, $\boldsymbol{G}^k$ is still integrated into $\boldsymbol{G}^A$ to form $\boldsymbol{G}^{A'}$ and could potentially be incorporated into $\boldsymbol{G}^R$ based on another boolean parameter denoted as $\boldsymbol{G}^{R'}$ *instead of* $\boldsymbol{G}^R$. The parameter $[\boldsymbol{G}^k]$ $k$ represents the value of $k$ used for executing kNN to derive $\boldsymbol{G}^k$. Similar notations apply to the determination of $k$ for performing kNNs to obtain $\boldsymbol{G}^L$, $\boldsymbol{G}^R$, and $\boldsymbol{G}^S$.

The parameters $[\boldsymbol{G}^L]$ *Freq.* and $[\boldsymbol{G}^S]$ *Freq.* denote the maximum frequency utilized in generating the relation graphs for LapPE and SignNet. The parameter $[\boldsymbol{G}^S]$ *Arch.* specifies the architecture employed to compute the SignNet PSEs, while $[\boldsymbol{G}^R]$ *Kernel* represents the specific kernel parameter used in computing RWSEs.

The hyperparameters $[\boldsymbol{G}^O]$ $K$, $\tau$, and $\epsilon$ correspond to the number of prototypes, softmax temperature in Eq. 4, and the entropy balancing factor in Eq. 12. Additionally, $[\boldsymbol{G}^O]$ $K_g/N$ denotes the proportion of $K_g$ to the $\boldsymbol{G}$ mini-batch size used in sparsifying the relation graph $\boldsymbol{G}^O$ within $F_K$. For instance, in the case of the Amazon Computers dataset, this parameter is set to 8, and the mini-batch size is 3072, resulting in $K_g = 8 \times 3,072 = 24,578$. The parameter $[\boldsymbol{G}^O]$ *S.K. #Iters.* represents the number of iterations for the Sinkhorn-Knopp algorithm, and $\alpha_1$ serves as the coefficient for the optimal transport alignment term $\mathcal{L}_O$.

### B.7.3 Hyperparameter Analysis

This section extends § 4.4 by delving into an analysis of selected hyperparameters, shedding light on their significance and some valuable insights gained from their impact on downstream performance and other metrics. Since node classification accuracy may not be a sufficiently sensitive metric and is confined to a specific task, we present values for our other predefined metrics outlined in § B.4. These evaluations are conducted explicitly on the Amazon Computers dataset while keeping all other hyperparameters fixed as per Table 6.

Fig. 9 represents a special scenario where $D_H = D_Z$. As we simultaneously alter the feature dimensions of representations and embeddings, we observe a corresponding adjustment in accuracy and the rank of these elements. This underscores the model's effectiveness in harnessing larger feature dimensions.

In Fig. 10, we explore the impact of the mini-batch size $N$ employed in constructing relation graphs. The consistent stability in accuracy underscores our model's ability to capture meaningful directions for guiding the invariance term, even with a reduced number of points at each iteration. However, there is a slight improvement in *corr* $\boldsymbol{H}$, *corr* $\boldsymbol{Z}$, and *nstd* $\boldsymbol{H}$ with larger mini-batch sizes. The observed degradation in *std* $\boldsymbol{Z}$ with larger mini-batch size may be attributed to utilizing the same number of iterations for all models,

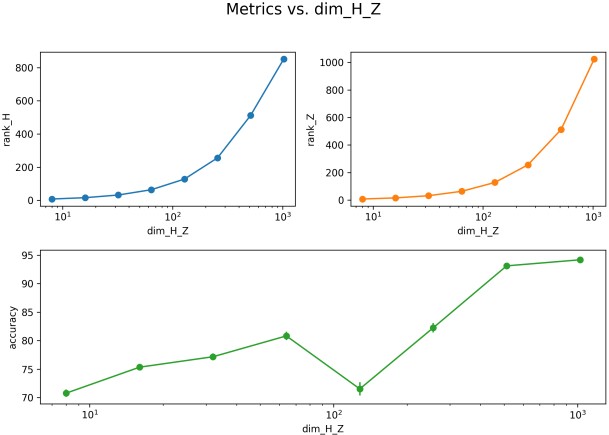

Figure 9: Impact of simultaneously altering $D_H$ and $D_Z$ in the particular case where $D_H = D_Z$ on our metrics and downstream accuracy for Amazon Computers.

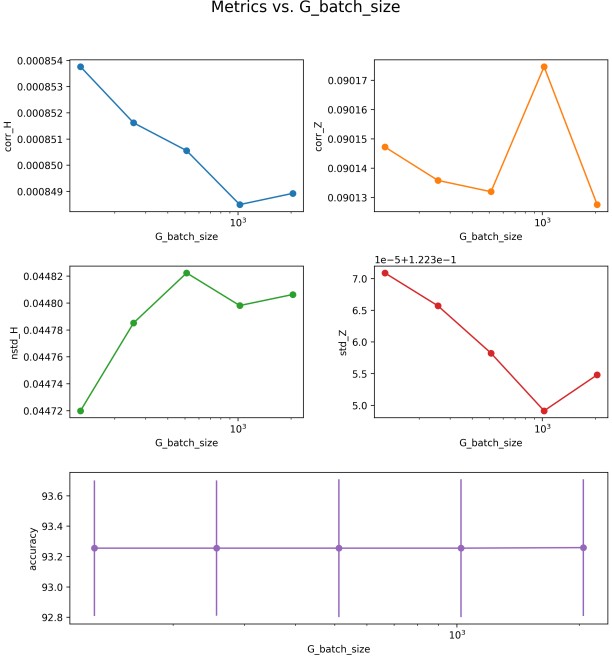

Figure 10: Impact of altering $N$, $\boldsymbol{G}$ mini-batch size, on our metrics and downstream accuracy for Amazon Computers.

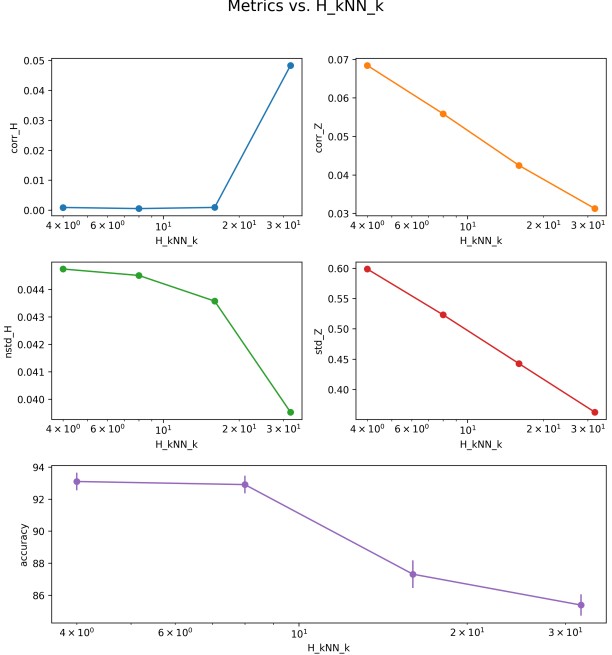

Figure 11: Impact of altering $[\boldsymbol{G}^k]$ $k$ on our metrics and downstream accuracy for Amazon Computers.

and this effect could potentially be mitigated with prolonged training, as the variance term $\mathcal{L}_V$ directly compares the spread with 1.

In Fig. 11, we investigate the impact of varying the parameter $k$ for kNN while utilizing a standalone $\boldsymbol{G}^k$. This illustration reveals that performance tends to drop as the number of points selected in kNN to guide the invariance loss $\mathcal{L}_{I'}$ becomes excessively large. This decline is attributed to the heightened noise introduced by abundant guidance points in the kNN process.

In Fig. 12, we explore the influence of altering the number of prototypes $K$ in the clustering module. The overall trend highlights the relative stability of downstream accuracy in relation to the choice of the number of clusters, a typically challenging task across different datasets. Moreover, the slightly increasing trend underscores the model's capability to leverage a higher number of prototypes. Therefore, there is no necessity for tedious tuning specific to each dataset; having a sufficiently large number of prototypes proves to be effective across diverse scenarios.

In Fig. 13, we explore the influence of varying $\alpha_1$, the coefficient for $\mathcal{L}_O$. As indicated by this figure and the ablation studies in Table 2, the model exhibits improved performance as long as this loss term is present, with a slightly more recognizable impact within a specific range of the coefficient $\alpha_1$. However, over-enforcement of this alignment leads to a worsening performance.

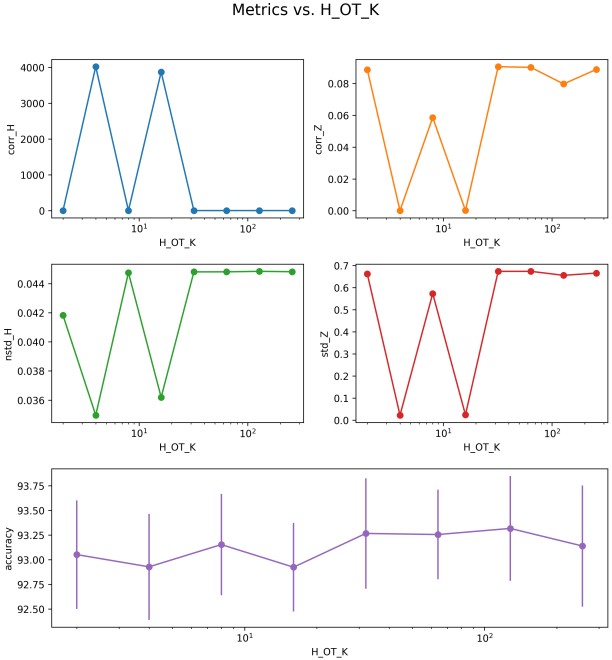

Figure 12: Impact of altering $[\boldsymbol{G}^O]$ $K$ indicating the number of prototypes on our metrics and downstream accuracy for Amazon Computers.

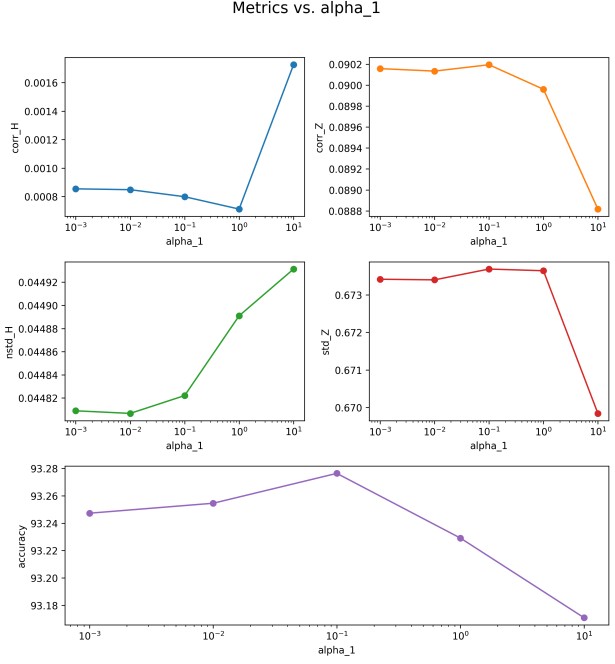

Figure 13: Impact of altering $\alpha_1[\mathcal{L}_O]$ indicating the coefficient for optimal transport alignment term on our metrics and downstream accuracy for Amazon Computers.

