# OpenReview forum: "ExGRG: Explicitly-Generated Relation Graph for Self-Supervised Representation Learning"
_TMLR — Rejected by TMLR_

### Review · Reviewer_NrfZ · 2024-08-01

**Summary Of Contributions:**

This work proposes a new strategy for self-supervised learning of node representations for graphs. Previous works have generally either (1) trained representations to be similar for views of the same node in different augmentations of the source graph, or (2) trained representations to be similar for similar but different nodes in the same graph. This work combines these approaches by building a weighted "relational graph", which includes both similarities between the same node in different augmented graph views and similarities between different nodes in the same augmented graph view. The authors describe a large number of strategies for constructing these relational graphs, and also propose using a hypernetwork to balance the different relational graph weights. They evaluate their representations using linear probing for node classification and find that it yields better representations than many previous baselines.

**Audience:**

Yes

**Broader Impact Concerns:**

No broader impact concerns.

**Claims And Evidence:**

Yes

**Requested Changes:**

I would like to see the following issues addressed before recommending acceptance:

### 1. Clarify the setting and contributions of this work in the intro and related work section

Although section 1 describes many of the components of the proposed model, and section 2 lists a large number of previous works, I found it difficult to understand the overall setting of the work and the scope of the proposed contributions beyond the previous approaches. I think sections 1 and 2 could be rewritten to more clearly state the problem setting and the key contributions.

Specific things I think would be worth stating more explicitly in Section 1:

- That ExGRG is primarily a method for learning node-level features, not graph-level features. This is somewhat ambiguous from Section 1, which refers to "data samples", "candidates", and "points" without clearly stating that these are the nodes of the original graph. (The appendix states that ExGRG could be extended to graph-level features, but the paper focuses primarily on node-level features.)
- That ExGRG uses *both* augmentations and additional relational information to construct the relation graph. In my first read, I interpreted Section 1 as saying that ExGRG used the relational information as a substitute for augmentations, and I was not sure how this work went beyond the augmentation-free methods AFGRL and SPGCL.

Also, I think the last paragraph of Section 2.2 could be rewritten to be more explicit about the relationship to previous augmentation-free approaches. It currently states "We distinguish ourselves by offering a non-contrastive comprehensive framework", but it's not clear what it means for it to be "comprehensive" or why previous work was not comprehensive. Additionally, statements like "marks the pioneering attempt to employ a learnable clustering mechanism to drive the invariance term, departing from relying solely on augmentation" seem to also apply to AFGRL, so I'm not sure "pioneering" is an appropriate term.

### 2. Explain how and when "intra-graph relations" are used

Based on Section 3, and in particular Sections 3.1.1 and 3.2.3, my understanding is that multiple augmented copies of the original graph are combined together into a single larger graph, and that this larger mini-batch graph is used to construct the explicit relation graphs for all of the following steps. As I understand it, this means that each relation graph can include both intra-view connections (scenario (ii)) and inter-view connections (scenarios (i) and (iii)).

However, section 4.3 includes an ablation for "Enforcing all relation graphs to consider intra-view relations alongside inter-view ones", and Appendix B.5 states that "the decision on whether to consider intra-relations for each G(i) is determined independently through balancing with mini-batch size considerations." I don't understand what this means, and I think this should be clarified. Is it not true that intra-graph relations are always considered in the relation graphs? What does it mean to not consider them, given that the relation graphs in Section 3 seem to all refer to the combined mini-batch graph as a whole?

### 3. Explain the "SG mechanism" explicitly

The paper makes multiple references to the "SG mechanism", but does not appear to ever explicitly define it or explain what it means. Section 3.2.2 just says SG can be employed, but not what it is. Section 3.4 says that $F\_{SG}$ "enforces the SG mechanism outlined in § 3.5", but Section 3.5 doesn't provide any more details, just referring back to Equation (7) in Section 3.4.

Does "SG mechanism" just mean "stop-gradient"? I don't think this is ever stated anywhere in Section 3, but from context it seems likely. (SG does seem to refer to stop-gradient in section 2.1, which discusses previous work. However, this was difficult to find on my first read.)

I would suggest explicitly writing out that $F\_{SG}$ is a stop-gradient operator, and explain what a stop-gradient operator does (e.g. that it prevents the flow of gradients and thus prevents optimization).

### 4. Expand on details for hyperparameter tuning and train/valid/test sets

The paper includes some details about hyperparameters in Section 4.5 and Appendices B.7 and B.3, but I don't fully understand how these hyperparameters were found. In particular, Appendix B.3 states that hyperparameters are evaluated based on classifier performance on a validation set, and that this process was repeated for 20 trials with different train/valid/test sets and aggregated to produce Table 1. However, Appendix B.7 indicates that the results in Table 1 for each dataset use a single set of optimal parameters, given in Table 6.

How does this process work? If hyperparameters were tuned based on each train/valid/test split, and there are 20 trials with 20 different splits, I would expect the selected hyperparameters to be somewhat dependent on which training and validation split was used. Did the 20 trials instead use the same hyperparameters, and if so, how were those values determined?

I think it would also be worth discussing how hyperparameters were tuned for the baseline approaches. Did they use the same tuning setup and the same 20 train/valid/test splits?

### Other comments and suggestions

I also have a few more minor questions and suggestions.

- Section 1: I'm not sure it makes sense to refer to the similarity measurements in the relation graph as "probabilistic", since many of them are not derived from any particular probability distribution and not normalized to represent a probabilistic quantity.
- Section 3.2.3: Would it be fair to say that the augmentation-free prior work, such as AFGRL and SPGRL, only support scenario (ii)? If so, I think mentioning this explicitly could help readers understand the contributions of this work better.
- Section 3.3.3: I think "Various methods have been introduced" or even "Various methods have been previously introduced" would be better phrasing than "Various methods are introduced", since this statement seems to be about previous work. Also, the idea of using clustering seems somewhat similar to the k-means approach of AFGRL, which might be worth mentioning.
- Section 3.4: I don't have a good intuition for why you need to learn the coefficients via a MLP here, instead of just treating them as hyperparameters. Section 4.3 mentions that this is important for avoiding collapse; is this because otherwise a specific relation graph ends up dominating the loss due to the relation graph depending circularly on the learned representations? I think it would be useful to give a bit more explanation of why these should be learned and what objective they are being optimized for. (Does this interact with the stop gradient?)

**Strengths And Weaknesses:**

**Strengths:**

- **S1.** The idea of using a single relation graph to unify relationships between augmented graph views and between nodes in the same view is interesting and seems to work better than using either alone.
- **S2.** The description of the method in section 3 is clearly written and explains many of the details of the proposed architecture (with a few exceptions detailed below).
- **S3.** The authors conduct thorough experiments and include a detailed ablation study, which provides additional information about the effect of each of the proposed components.

**Weaknesses:**

- **W1.** I found the introduction and related work sections to be somewhat hard to follow, and I think these sections are somewhat vague about the actual contributions of the paper beyond previous work.
- **W2.** A few details of the proposed approach are not clearly explained. Specifically, details are missing about the "SG mechanism", and the description of "intra-view relations" in Section 3.2.3 seems incomplete based on later discussion in the ablation studies and appendix. Additionally, there are a few parts of Section 3 that could refer more directly to previous work.
- **W3.** The proposed approach seems to have a large number of hyperparameters, which might make it difficult to apply effectively in practice. Relatedly, I think some details are missing regarding how the hyperparameters were tuned, which seems important for evaluating the results fairly.
- **W4.** The stated connections to expectation-maximization and Laplacian eigenmaps seem a bit weak and informal, and I'm not sure they provide much additional motivation for the proposed method.

---

> ### Author Response · Authors · 2024-09-14
> **Appreciation for Your Valuable Feedback**
>
> We appreciate the time and effort you have dedicated to providing this valuable feedback. In the following, we address each of the concerns and questions you have raised.

---

> > ### Comment · Reviewer_NrfZ · 2024-09-20
> > **Discussion**
> >
> > I am glad you found my feedback useful and I believe your planned revisions will improve the paper.
> >
> > In your responses, you mention a number of changes you plan to make in the final revision. Are you aware that TMLR allows you to upload revised versions of the manuscript during the review process? I would find it useful to see a revised version of the submission before submitting the final recommendation, especially regarding the clarity issues that I discussed in my review.
> >
> > Also, you discuss removing details due to space limitations. TMLR [does not impose hard limitations on the length of a submission](https://jmlr.org/tmlr/author-guide.html) (as long as the length is "justified by its content"), so I'd encourage you to add back details as needed to ensure that the method is explained fully.
> >
> > Some more specific responses:
> >
> > **W3.1:** Although not a big concern, I'm not fully convinced by your response about the number of hyperparameters, and I still think it might be tricky to tune them well.
> >
> > - Figure 11 seems to show a substantial sensitivity of the accuracy to the value of $[G^k]~k$. Also, this plot is only for a single dataset, Amazon Computers. So I don't think this plot is enough to determine that "the suitable range for searching and selecting each of these HPs ... remains constrained and relatively consistent across different datasets.".
> > - From table 6, it looks like about 40% (16/40) of the hyperparameters are new and specific to your method, right?
> > - I also don't agree that Table 6 shows that they are "the mentioned remaining HPs are typically in specific ranges and maintain close similarity across diverse datasets". Each of the $k$ parameters, for instance, spans the range from 4 to 32, and different values of $k$ seem to be necessary both across different datasets and between the different relation graph types. As another example, the $\alpha\_1[LO]$ parameter ranges from 0.01 to 1.
> >
> > **W3.2 and Ch4:** Thanks for your response. It would be great if you could add these details to the appendix.
> >
> > Also, I'm still a bit confused about how the different dataset splits interact. If I understand correctly, you first use one train/valid/test set to tune the hyperparameters of the pretraining step (by training a classifier on the training set and using the validation set to tune), and this is what is shown in Table 6. You then freeze the pretraining hyperparameters, create 20 more train/valid/test sets for the same dataset, and then separately tune the classifier hyperparameters on those (with the results shown in Table 1)?
> >
> > Doesn't this open up potential for leakage between the original training/valid set (used to find pretraining hyperparameters) and the final test sets? If hyperparameters were tuned to do well on the points in the original validation set, and then you re-split those points to use them in a test set, the results in Table 1 might be unfairly high.
> >
> > Or, am I misunderstanding this and you use the same test set for both steps, and only the train/valid split is re-randomized in Table 1?
> >
> > **Ch2.2:** Thank you for clarifying. So, in Table 4, in the "Intra Relations" setting every individual $G^{(i)}$ includes intra-view relations, but in the main results (e.g. Table 1) some of the $G^{(i)}$ include these relations and others do not?
> >
> > I'm glad you are planning to add some details about this in the paper. Currently, I don't think it is discussed anywhere that you disable intra-view relations for a subset of the of the graphs by default. (Is it just Neighborhood Similarity or are there others?)
> >
> > **M1 - M4:** Thanks for the additional information, your responses and planned changes make sense to me.

---

> ### Author Response · Authors · 2024-09-14
> **W1**
>
> **W1. I found the introduction and related work sections to be somewhat hard to follow, and I think these sections are somewhat vague about the actual contributions of the paper beyond previous work.**
>
> Thank you for the insightful suggestions and feedback. We will revise these two sections in the final version to emphasize our contributions more clearly. In the current submission, we present an overview of the primary contributions in Section 1, paragraph 3, with further elaboration provided in paragraphs 4 and 5. Moreover, we address the distinction of our work in Section 2, paragraph 2.

---

> ### Author Response · Authors · 2024-09-14
> **W2.1**
>
> **W2.1. A few details of the proposed approach are not clearly explained. Specifically, details are missing about the "SG mechanism", and the description of "intra-view relations" in Section 3.2.3 seems incomplete based on later discussion in the ablation studies and appendix.**
>
> Intra-view Relations: We have made considerable efforts to emphasize our model's ability to capture both inter-view and intra-view relations by aggregating both views into a unified notation, as introduced in Section 3.1.1. This capability is further discussed explicitly in Section 3.2.3, paragraph 1. Additional details, constrained by space limitations, are provided in the final paragraph of Section B.5.
>
> SG Mechanism: Although we implement the SG mechanism in a different part of the architecture (as shown in Fig. 1 and Eq. 7) compared to previous approaches, this mechanism itself is not novel, having been explored in prior works [1]. Therefore, we have included its details in Appendix A.7.1.
>
> [1] Grill, Jean-Bastien, Florian Strub, Florent Altché, Corentin Tallec, Pierre Richemond, Elena Buchatskaya, Carl Doersch et al. "Bootstrap your own latent-a new approach to self-supervised learning." Advances in neural information processing systems 33 (2020): 21271-21284.

---

> ### Author Response · Authors · 2024-09-14
> **W2.2**
>
> **W2.2. Additionally, there are a few parts of Section 3 that could refer more directly to previous work.**
>
> We have made every effort to reference prior works whenever relevant ideas were adopted. However, we welcome any additional details that could help us address any references or points we may have unintentionally overlooked in the literature.

---

> ### Author Response · Authors · 2024-09-14
> **W3.1**
>
> **W3.1. The proposed approach seems to have a large number of hyperparameters, which might make it difficult to apply effectively in practice.**
>
> Like other SSL methods and deep learning approaches in general, the five coefficients governing the loss terms in Eq. 10, along with other hyperparameters (HPs), such as the number of prototypes or k in kNNs, as listed in Table 6, necessitate tuning. However, our investigations reveal that the suitable range for searching and selecting each of these HPs, based on prior literature and empirical observations, remains constrained and relatively consistent across different datasets. As depicted in Fig. 11-13, we illustrate the stability of downstream performance across a spectrum of selected HP values. This observation suggests that the overall performance is not excessively sensitive to the precise choice of HP values; thus, a broad acceptable range exists instead.
>
> Furthermore, HPs in Table 6 are mostly common among SOTA SSL methods. These include batch size, optimizer, learning rate, number of iterations, encoder and expander/projector activations, normalizations, number of layers, hidden/output dimensions, and augmentation parameters for nodes/edges within each view. Consequently, our model requires tuning only a few remaining HPs in addition to other SSL methods. Table 6 shows that the mentioned remaining HPs are typically in specific ranges and maintain close similarity across diverse datasets. Even for this limited number of HPs, the narrow scope for parameter search is primarily influenced by previous works leveraging deep clustering in other applications [2, 3] or PSEs for graph transformers [4, 5].
>
> [2] Asano, Yuki Markus, Christian Rupprecht, and Andrea Vedaldi. "Self-labelling via simultaneous clustering and representation learning." arXiv preprint arXiv:1911.05371 (2019).
>
> [3] Caron, Mathilde, Ishan Misra, Julien Mairal, Priya Goyal, Piotr Bojanowski, and Armand Joulin. "Unsupervised learning of visual features by contrasting cluster assignments." Advances in neural information processing systems 33 (2020): 9912-9924.
>
> [4] Cantürk, Semih, Renming Liu, Olivier Lapointe-Gagné, Vincent Létourneau, Guy Wolf, Dominique Beaini, and Ladislav Rampášek. "Graph positional and structural encoder." arXiv preprint arXiv:2307.07107 (2023).
>
> [5] Rampášek, Ladislav, Michael Galkin, Vijay Prakash Dwivedi, Anh Tuan Luu, Guy Wolf, and Dominique Beaini. "Recipe for a general, powerful, scalable graph transformer." Advances in Neural Information Processing Systems 35 (2022): 14501-14515.

---

> ### Author Response · Authors · 2024-09-14
> **W3.2**
>
> **W3.2. Relatedly, I think some details are missing regarding how the hyperparameters were tuned, which seems important for evaluating the results fairly.**
>
> Our hyperparameter tuning process follows a similar approach to prior graph SSL methods, particularly BGRL [6]. To identify optimal hyperparameters, we use a single train/validation/test partition for each dataset. We leverage the Bayesian Optimization algorithm in Ray Tune to efficiently conduct hyperparameter searches for each dataset. Consistent with standard practice, the validation split is used to select hyperparameters that maximize downstream performance for that specific validation split. The final performance results presented in the paper, such as those in Table 1, are reported based on the previously selected hyperparameters, evaluated on the test splits across 20 random model initializations and dataset partitions.
>
> [6] Thakoor, Shantanu, Corentin Tallec, Mohammad Gheshlaghi Azar, Mehdi Azabou, Eva L. Dyer, Remi Munos, Petar Veličković, and Michal Valko. "Large-scale representation learning on graphs via bootstrapping." arXiv preprint arXiv:2102.06514 (2021).

---

> ### Author Response · Authors · 2024-09-14
> **W4**
>
> **W4. The stated connections to expectation-maximization and Laplacian eigenmaps seem a bit weak and informal, and I'm not sure they provide much additional motivation for the proposed method.**
>
> In Sections 3.2.1 and 3.2.2, we draw theoretical connections between our proposed model and established methods. These perspectives link our algorithm to the Laplacian Eigenmap (LE) and Expectation Maximization (EM). Our analysis builds on prior work that connects SSL algorithms with spectral embedding techniques in manifold learning [2] and the implicit EM optimization framework for non-contrastive SSL [3].
>
> By introducing ExGRG, we address the issue of Laplacian rank deficiency, while also making the alternating steps in the EM perspective explicit. These theoretical connections provide deeper insights into why the method works, rather than treating it as merely heuristic or ad hoc. Additionally, they highlight potential avenues for improvement by leveraging well-explored algorithms from the literature.
>
> [2] Balestriero, Randall, and Yann LeCun. "Contrastive and non-contrastive self-supervised learning recover global and local spectral embedding methods." Advances in Neural Information Processing Systems 35 (2022): 26671-26685.
>
> [3] Chen, Xinlei, and Kaiming He. "Exploring simple siamese representation learning." In Proceedings of the IEEE/CVF conference on computer vision and pattern recognition, pp. 15750-15758. 2021.

---

> ### Author Response · Authors · 2024-09-14
> **Ch1.1**
>
> **Ch1.1. Clarify the setting and contributions of this work in the intro and related work section. Although section 1 describes many of the proposed model's components and section 2 lists many previous works, I found it difficult to understand the overall setting of the work and the scope of the proposed contributions beyond the previous approaches. I think sections 1 and 2 could be rewritten to more clearly state the problem setting and the key contributions.**
>
> Thank you for your feedback. We will restructure these two sections to more clearly articulate our contributions, as well as highlight the similarities and distinctions between our approach and previous methods more explicitly.

---

> ### Author Response · Authors · 2024-09-14
> **Ch1.2**
>
> **Ch1.2. That ExGRG uses both augmentations and additional relational information to construct the relation graph. In my first read, I interpreted Section 1 as saying that ExGRG used the relational information as a substitute for augmentations, and I was not sure how this work went beyond the augmentation-free methods AFGRL and SPGCL.**
>
> Thank you for the valuable feedback. We will address this in the final version. While we used phrases like "instead of relying solely on the conventional augmentation-based...", we acknowledge the need to clarify more explicitly in Section 1 that our approach still incorporates augmentation.

---

> ### Author Response · Authors · 2024-09-14
> **Ch1.3**
>
> **Ch1.3. Also, I think the last paragraph of Section 2.2 could be rewritten to be more explicit about the relationship to previous augmentation-free approaches. It currently states, "We distinguish ourselves by offering a non-contrastive comprehensive framework", but it's not clear what it means for it to be "comprehensive" or why previous work was not comprehensive. Additionally, statements like "marks the pioneering attempt to employ a learnable clustering mechanism to drive the invariance term, departing from relying solely on augmentation" seem to also apply to AFGRL, so I'm not sure "pioneering" is an appropriate term.**
>
> Thank you for your valuable feedback. Based on your suggestions, we will revise these two sections to convey our intent more clearly.
>
> The term "comprehensive" refers to the core framework of our algorithm. Specifically, it highlights the possibility of exploring strategies beyond those described in Section 3.3, depending on the data domain and context, to construct explicit relational graphs. In contrast, other components of the algorithm remain unchanged. Additionally, it refers to our approach for aggregating guidance from multiple sources (as described in Section 3.4), including those outlined in Section 3.3 and through augmentation.
>
> The term "pioneering" primarily refers to the "learnable clustering mechanism" in our model, which contrasts with KMeans, the non-learnable approach employed in AFGRL. Our method introduces a clustering module that is trained end-to-end alongside the encoder, following an EM-style optimization where both deep components co-evolve, reflecting a dynamic interplay. Moreover, "pioneering" signifies one of the critical distinctions of our approach, which uses a learnable clustering algorithm for representation learning by guiding invariance, setting it apart from previous methods. This is discussed in greater detail in the first three paragraphs of Section 3.3.3.

---

> ### Author Response · Authors · 2024-09-14
> **Ch2.1**
>
> **Ch2.1. Explain how and when "intra-graph relations" are used. Based on Section 3, and in particular Sections 3.1.1 and 3.2.3, my understanding is that multiple augmented copies of the original graph are combined together into a single larger graph, and that this larger mini-batch graph is used to construct the explicit relation graphs for all of the following steps. As I understand it, this means that each relation graph can include both intra-view connections (scenario (ii)) and inter-view connections (scenarios (i) and (iii)).**
>
> Thank you for the comment. Your observation is correct. The relation graphs are indeed constructed based on what you referred to as a "larger mini-batch," which is precisely why the notations in Section 3.1.1 are defined as such. Specifically, in augmentation-based relation graphs, only a subset of the inter-view entries are non-zero. However, as we discuss in Section 3.2.3, our more general approach to constructing relation graphs permits both inter-view and intra-view relations to be represented, thereby encompassing a broader range of potential interactions.

---

> ### Author Response · Authors · 2024-09-14
> **Ch2.2**
>
> **Ch2.2. However, section 4.3 includes an ablation for "Enforcing all relation graphs to consider intra-view relations alongside inter-view ones", and Appendix B.5 states that "the decision on whether to consider intra-relations for each G(i) is determined independently through balancing with mini-batch size considerations." I don't understand what this means, and I think this should be clarified. Is it not true that intra-graph relations are always considered in the relation graphs? What does it mean to not consider them, given that the relation graphs in Section 3 seem to all refer to the combined mini-batch graph as a whole?**
>
> Thank you for your thorough review and attention to detail. You are correct. The explanation in the current manuscript was condensed to adhere to page limitations, but we will provide further clarification in the final version.
>
> As outlined in Appendix B.5, our ablation studies indicate that enforcing intra-view relations across all relation graphs can potentially yield improved representations, albeit with added computational overhead. The compositional relation graph $G$ (Eq. 7) consistently incorporates some intra-view relations. However, for a particular individual graph $G^{(i)}$ in Eq. 7, the Neighborhood Similarity graph, our empirical results suggest that considering only inter-view relations is sufficient to achieve comparable performance without the added computational burden of accounting for all relations, including intra-view ones.
>
> In this scenario, the remaining relation graphs $G^{(i)}$ in Eq. 7, along with the compositional graph $G$, still contain nonzero intra-view relations. Our hypothesis for why inter-view connections alone are effective in the Neighborhood Similarity relation graph is as follows: we believe that the k-nearest neighbors (kNN) identified in the other view (inter-view) for each anchor node already provide sufficiently strong neighboring relations. Thus, intra-view neighbors are not necessary to guide the invariance term.

---

> ### Author Response · Authors · 2024-09-14
> **Ch3**
>
> **Ch3. Explain the "SG mechanism" explicitly. The paper makes multiple references to the "SG mechanism", but does not appear to ever explicitly define it or explain what it means. Section 3.2.2 just says SG can be employed, but not what it is. Section 3.4 says that $F_{SG}$ "enforces the SG mechanism outlined in § 3.5", but Section 3.5 doesn't provide any more details, just referring back to Equation (7) in Section 3.**
>
> Thank you for your feedback. You are correct, and we will make the definition of the stop-gradient (SG) mechanism more explicit in the final version, as per your suggestion. In Eq. 7, $F_{SG}$ represents the stop-gradient operation, which functions as an identity during the forward pass but ensures zero gradients are propagated during the backward pass. We appreciate your input and will clarify this point more directly to avoid any ambiguity.

---

> ### Author Response · Authors · 2024-09-14
> **Ch4**
>
> **Ch4. Expand on details for hyperparameter tuning and train/valid/test sets
> The paper includes some details about hyperparameters in Section 4.5 and Appendices B.7 and B.3, but I don't fully understand how these hyperparameters were found. In particular, Appendix B.3 states that hyperparameters are evaluated based on classifier performance on a validation set and that this process was repeated for 20 trials with different train/valid/test sets and aggregated to produce Table 1. However, Appendix B.7 indicates that the results in Table 1 for each dataset use a single set of optimal parameters, given in Table 6. How does this process work? If hyperparameters were tuned based on each train/valid/test split, and there are 20 trials with 20 different splits, I would expect the selected hyperparameters to be somewhat dependent on which training and validation split was used. Did the 20 trials instead use the same hyperparameters, and if so, how were those values determined? I think it would also be worth discussing how hyperparameters were tuned for the baseline approaches. Did they use the same tuning setup and the same 20 train/valid/test splits?**
>
> Thank you once again for your detailed feedback and careful consideration of our work. Based on your input, we will explicitly address this concern in the final version. For further details on hyperparameter (HP) optimization, please refer to our response to W3.2.
>
> In the context of SSL, representation learning and evaluation typically occur in two stages: pre-training and training. The "pre-training" phase refers to learning the encoder and expander weights without the use of labels. Conversely, in the "training" phase, the expander is replaced by a classifier, and the classifier is trained for the downstream task while keeping the encoder's weights fixed. SSL algorithms, including ExGRG, focus primarily on the pre-training phase. There are two distinct types of HPs involved in this process. The first type pertains to pre-training (as shown in Table 6), while the second type is associated with training, such as the HPs for the L2-regularized logistic regression classifier. The term "hyperparameter" in Section B.3, to which you referred, pertains to the latter.
>
> Our methodology follows the standard experimental setup used in graph SSL literature and baselines, which we clarify as follows: To determine the pre-training HPs (Table 6), we employ a fixed random train/validation/test partitioning and use the validation split for HP tuning. A classifier is trained on the validation split to assess the performance of each candidate set of pre-training HPs. The classifier's training HPs are optimized to achieve the best downstream accuracy with the selected pre-training HPs. In summary, the process consists of searching for an optimal set of pre-training HPs, performing the pre-training step, and then training the best downstream classifier by optimizing its training HPs on the validation split. We report validation performance for each candidate set of pre-training HPs before moving on to the next candidate.
>
> To ensure a fair comparison with the baselines (Table 1), we follow the same approach used in the existing graph SSL literature. The pre-training HPs (Table 6) remain fixed, as determined previously. The pre-training step is performed once, after which the encoder's weights are frozen. We then conduct the downstream training step over 20 trials to account for performance variability due to different partitioning in node classification datasets. In each trial, we optimize the training HPs on the validation split and subsequently report the test performance.

---

> ### Author Response · Authors · 2024-09-14
> **M1**
>
> **M1. Section 1: I'm not sure it makes sense to refer to the similarity measurements in the relation graph as "probabilistic", since many of them are not derived from any particular probability distribution and not normalized to represent a probabilistic quantity.**
>
> We will address this in the final version. Specifically, during the construction of the relation graphs, all entries are normalized to lie within the range of 0 to 1.

---

> ### Author Response · Authors · 2024-09-14
> **M2**
>
> **M2. Section 3.2.3: Would it be fair to say that the augmentation-free prior work, such as AFGRL and SPGRL, only support scenario (ii)? If so, I think mentioning this explicitly could help readers understand the contributions of this work better.**
>
> Thank you for the suggestion; it's a great idea, and we will consider incorporating it into the manuscript. To summarize, ExGRG supports all three scenarios. Augmentation-based methods, in contrast, only support scenario (i). SPGCL, being an augmentation-free method with a single view, only involves inter-view relations that align with scenario (ii). AFGRL, another augmentation-free method, constructs two views via the EMA encoder, which restricts it to scenario (iii).

---

> ### Author Response · Authors · 2024-09-14
> **M3**
>
> **M3. Section 3.3.3: I think "Various methods have been introduced" or even "Various methods have been previously introduced" would be better phrasing than "Various methods are introduced", since this statement seems to be about previous work. Also, the idea of using clustering seems somewhat similar to the k-means approach of AFGRL, which might be worth mentioning.**
>
> We will revise this section based on your input. Thank you for your feedback.

---

> ### Author Response · Authors · 2024-09-14
> **M4.1**
>
> **M4.1. Section 3.4: I don't have a good intuition for why you need to learn the coefficients via a MLP here, instead of just treating them as hyperparameters.**
>
> As stated in Section 3.4, the coefficients generated by the MLP in Eq. 8 determine the contribution of each relation graph to the final compositional relation graph (Eq. 7). Our findings indicate that using the proposed hypernetwork MLP to dynamically adjust these contributions during pre-training, based on the current characteristics of each relation graph, is more effective than relying on a fixed contribution value determined by a hyperparameter. This online approach better adapts to the varying nature of the relation graphs, leading to improved performance.

---

> ### Author Response · Authors · 2024-09-14
> **M4.2**
>
> **M4.2. Section 4.3 mentions that this is important for avoiding collapse; is this because otherwise a specific relation graph ends up dominating the loss due to the relation graph depending circularly on the learned representations? I think it would be useful to give a bit more explanation of why these should be learned and what objective they are being optimized for. (Does this interact with the stop gradient?)**
>
> As you correctly pointed out, using the hyperparameter (HP) method can result in dimensional collapse in some instances, which is discussed in point (iii) of the Ablation Studies in Section 4.3. Additionally, there is no interaction with the Stop-Gradient mechanism, as evidenced by tracing the gradient paths through the equations or the architecture illustrated in Fig. 1.
>
> Relying on fixed HPs to determine the contribution of each relation graph to the final compositional relation graph fails to account for the evolving characteristics of the contributing relation graphs during training. In contrast, our approach ensures that all learnable components, including these coefficients, are trained end-to-end using the objective function defined in Eq. 10.

---

> ### Author Response · Authors · 2024-09-27
> **Reply to "Discussion"**
>
> We sincerely appreciate your prompt response and the thoughtful attention you have given to our work. We are dedicated to addressing your concerns and offering additional clarification where needed. We have also uploaded a revised manuscript and included detailed discussions of the specific changes made in response to feedbacks.
>
>
> **W3.1:**
>
> We acknowledge that referencing Fig. 11 to demonstrate performance stability for varied hyperparameter (HP) values was an oversight. While Fig. 12-13 effectively illustrate stability across varying HPs (with all horizontal axes being logarithmic), Fig. 11, as noted in Section B.7.3, reveals a different phenomenon. Specifically, performance degrades as the number of points selected in kNN to guide the invariance loss becomes excessively large. This decline is likely due to the increased noise from the large number of guidance points used in the kNN process.
>
> The 40% mentioned is correct. Roughly half of the HPs in Table 6 are specific to our model, while the remaining HPs are common across typical graph SSL models, which may also have their own set of HPs. Although we significantly outperform the SOTA in Table 1, we do not claim that our HP tuning process is more efficient than that of previous models. Rather, our primary focus is on improving the robustness of the learned representations for downstream tasks.
>
> In Table 6, while certain HPs exhibit more variation across datasets, for instance, those with search spaces like {0.01, 0.02, 0.2, 0.5, 1} or {4, 8, 12, 16, 32, 48, 64, 80}, other HPs display considerably less variation.
>
>
>
> **W3.2 and Ch4:**
>
> Your understanding of our HP tuning process for both the pre-training and training phases, as detailed in our response to Ch4, is correct. In the original SSL approaches for the vision domain, no 20-trial setup exists. However, in graph SSL experimental setups, different data partitioning leads to performance variability, which is reflected in the standard deviation values shown in Table 1 for node classification datasets. To account for this, the results are averaged over 20 random model initializations and train/validation/test splits.
>
> The HP tuning approach we employed follows the standard practice in graph SSL. All SSL methods listed in Table 1 utilize the same approach, ensuring that the comparison remains fair. Additionally, this method is consistent with the approach you discussed latter, eliminating any potential for leakage.
>
>
> **Ch2.2:**
>
> As outlined in the paper, the first row in the ablation studies (Tables 2, 4, and 5) corresponds directly to the results reported in Table 1. The second row (only in Tables 2 and 4) represents the same configuration as the first row but with a reduced iteration count of 5k, providing a reference for the subsequent rows, which are also conducted with 5k iterations. The final row (Intra-Relations) modifies $G^k$ to consider intra-relationships. However, as described in Section 2.2, in the rows above, $G^k$ only captures inter-relations. We applied this approach specifically to Neighborhood Similarity $G^k$. We believe that the k-nearest neighbors identified from the other view (inter-view) for each anchor node already provide sufficiently strong neighboring relationships. Therefore, we avoid the intra-view neighbors and minimize additional computational overhead.
>
> The only mention of this distinction appeared in the last sentence of Section B.5. Based on your suggestion, we expanded on this explanation in the revised version to prevent further confusion. Now, this can be found in Section B.5.1.

---

> ### Author Response · Authors · 2024-09-27
> **Revision**
>
> **Ch1 (Section 1 & Section 2):** We have revised Section 1 and Section 2.2 to more explicitly and clearly emphasize our primary contributions beyond previous work. The main contribution of this research is now clearly highlighted in Section 1 (Paragraph 3). Additionally, we draw distinctions between our approach and prior methods more clearly in Section 2.2 (Paragraph 2).
>
> In particular, Section 1 now explicitly states that, in addition to addressing problematic augmentation, we introduce supplementary cues to refine invariance. In Section 2.2, we added: "Prior approaches either heavily rely on augmentation or discard it entirely. We distinguish our work by proposing a non-contrastive, comprehensive framework. Unlike previous methods, ExGRG incorporates graph augmentations by dynamically adjusting them through the generation and integration of additional guidance in the form of relation graphs."
>
> **W2.1 and Ch3 (SG):** Regarding the SG Mechanism, an explicit definition has been included in Section 3.4, accompanied by its corresponding mathematical notation.
>
> **W2.1 and Ch2 (Intra-view):** In response to comments on intra-view relations, links have been added in Section 3.2.3 to direct the reader to more detailed sections within the manuscript. A new section has been added to the appendix, Section B.5.1, to further address the concerns and provide additional clarification.
>
> **W3.2 and Ch4 (HP):** To address concerns regarding hyperparameter tuning, a new Section B.7.1 has been added in the appendix. This section provides a detailed explanation of our hyperparameter tuning procedure, aligning with standard practices in graph SSL.
>
> **M1:** We have removed the term "probabilistic" to avoid confusion.
>
> **M3:** The issue has been fixed, and the following statement has been added for clarity: "This marks a novel attempt to integrate a learnable clustering algorithm for representation learning, inspired by a non-learnable clustering approach, KMeans [1]."
>
> [1] Lee, Namkyeong, Junseok Lee, and Chanyoung Park. "Augmentation-free self-supervised learning on graphs." In Proceedings of the AAAI conference on artificial intelligence, vol. 36, no. 7, pp. 7372-7380. 2022.

---

> > ### Comment · Reviewer_NrfZ · 2024-09-30
> > **Comments on revision**
> >
> > Thanks for your reply and for uploading a revised version of the paper. A few remaining comments:
> >
> > ## W1 / Ch1 (Contributions)
> > The revised paper states the key contributions a bit more clearly, but I still think it is worth emphasizing a few aspects more:
> >
> > - I think the paper would benefit from clearly stating somewhere in the introduction that it focuses on learning node-level features.
> > - I also think it would be nice to be more explicit about the fact that augmentations are still being used. Perhaps instead of "our primary contribution is to integrate additional cues to more effectively guide the determination ...", you could say something like "our primary contribution is to combine graph augmentations with additional cues to more effectively guide the determination ..."?
> >
> > ## W2.1 / Ch2 (Intra-relations)
> >
> > Thanks for adding more details about the intra-relations in B.5.1.
> >
> > In your other response you say "However, as described in Section 2.2, in the rows above, $G^k$ only captures inter-relations." Where is this? I don't see any mention of $G^k$ in Section 2.2. I also would have expected to see this mentioned in Section 3.3.1 (the place where $G^k$ is defined), but I don't see anything there either.
> >
> > ## W2.1 / Ch3 (SG mechanism)
> >
> > Thanks for adding the extra detail about the SG mechanism. Could you also expand out the abbreviation in 3.4 to remind readers that this means "stop gradient"? So, instead of "enforces the SG mechanism", maybe "enforces the stop-gradient (SG) mechanism"?
> >
> > ## W3.2 /  Ch4 (Hyperparams)
> >
> > I'm still a bit confused by your description of the HP tuning process in B.7.1 and in your reply. What do you mean by "Additionally, this method is consistent with the approach you discussed latter, eliminating any potential for leakage."? Are you saying you do use the same test set for both steps?
> >
> > If so, please modify section B.7.1 to state this explicitly. For instance:
> > - For the sentence "Final performance metrics reported in Table 1 are based on evaluations conducted on the test split, averaged across 20 random model initializations and dataset partitions.", you could say something like "Final performance metrics reported in Table 1 are based on evaluations conducted on the fixed test split, averaged across 20 random model initializations and choices of the training and validation splits."
> > - For the sentence "We then conduct the downstream training step over 20 trials to account for performance variability due to different partitioning in node classification datasets. In each trial, we optimize the training hyperparameters on the validation
> > split and subsequently report the test performance.", you could say something like "We then conduct the downstream training step over 20 trials with different model initializations and different train/validation splits, to account for performance variability due to different partitioning in node classification datasets. In each trial, we optimize the training hyperparameters on the (trial-specific) validation split and subsequently report the performance on the (fixed) test set."
> >
> > ## Other minor comments
> >
> > - In Section 3.3.3, "non-learnable clustering approach, KMeans Lee et al. (2022)." should probably be formatted as "non-learnable clustering approach, KMeans (Lee et al. 2022)."

---

> > > ### Author Response · Authors · 2024-09-30
> > > **Revision**
> > >
> > > Thank you for your valuable feedback and the time you dedicated to reviewing our manuscript. Your insights have been essential in guiding us to improve our work. Below, we outline the specific changes made in response to your comments:
> > >
> > > **Ch1:**
> > > We have added the following clarification to the last paragraph:
> > > "Our proposed approach focuses on a node-level representation learning task."
> > >
> > > Additionally, we have revised the discussion on graph augmentations:
> > > "Given the inadequacy and misguidance of graph augmentations, our primary contribution lies in integrating augmentations with supplementary cues..."
> > >
> > > **Ch2:**
> > > Apologies for the incorrect reference in our previous response. The mention of *"Section 2.2"* was a typo. Based on your suggestion, we have now added a new paragraph explicitly discussing this matter in Section 3.3.1.
> > >
> > > **Ch3:**
> > > The part has been revised according to your suggestion.
> > >
> > > **Ch4:**
> > > This part has been modified as per your recommendation.
> > >
> > > **Minor comment:**
> > > This has been updated as suggested.
> > >
> > > We appreciate your detailed review and believe these revisions have strengthened the manuscript.

---

### Review · Reviewer_cKEx · 2024-08-03

**Summary Of Contributions:**

This work studies the self-supervised graph representation learning. The authors propose a new non-contrastive approach to Explicitly Generate a compositional Relation Graph (ExGRG) instead of relying solely on the conventional augmentation-based implicit relation graph. They argue that incorporating prior domain knowledge and online extracted information into the SSL invariance objective may avoid the misleading augmentation for contrastive learning. They further provide some discussion from the perspectives of the Laplacian Eigenmap (LE) perspective and the ExpectationMaximization (EM) to justify ExGRG. They also conduct extensive experiments to verify the effectiveness of ExGRG.

**Audience:**

Yes

**Claims And Evidence:**

No

**Requested Changes:**

1. The incorporated prior information may not be applicable to all graph SSL scenarios, for example, heterophilic graphs, or graph-level classification. In fact, the prior information can be directly incorporated into supervised learning to quickly learn the desired representations instead of taking such a complicated way;

2. It's unclear whether the prior information indeed resolves the problem.
- For the analysis based on LE, why original augmentation-based approaches can be considered as an LE optimization, if they are not using VICReg?
- Moreover, how does the EM-based analysis imply the advantages of ExGRG over previous augmentation-based approaches?

3. There are no experiments evaluating the hyperparameter sensitivity in terms of the objective Eq 10;


4. The accuracies in Cora and CiteSeer are significantly higher than those of the supervised approaches. It's suggested that the code and reproducibility be checked by the reviewers.

**Strengths And Weaknesses:**

**Strengths**

- How to tackle the misleading augmentation in SSL on graphs is an important problem;
- Incorporating some prior knowledge to construct the relation graph is an interesting idea to tackle the issue;
- The empirical improvements are quite impressive;

**Weakness**

- The incorporated prior information may not be applicable to all graph SSL scenarios;
- It's unclear whether the prior information indeed resolves the problem;
- There are no experiments evaluating the hyperparameter sensitivity in terms of the objective Eq 10;
- The accuracies in Cora and CiteSeer are significantly higher than those of the supervised approaches. It's suggested to check the code and reproducibility;

---

> ### Author Response · Authors · 2024-09-14
> **Appreciation for Your Valuable Feedback**
>
> We appreciate the time and effort you have taken to provide your valuable feedback. In the following sections, we address each of the concerns and questions you have raised.

---

> ### Author Response · Authors · 2024-09-14
> **W1**
>
> **W1. The incorporated prior information may not be applicable to all graph SSL scenarios;**
>
> The datasets used to evaluate ExGRG are directly sourced from prior work on Graph SSL, as detailed in Section 4.1. For these graph types, constructing relation graphs based on prior information formulated through Higher-Order Graph Encodings (Section 3.3.2) has been shown to be advantageous in our work. These encodings include positional and structural encoding methods, such as LapPE, RWSE, and SignNet. However, for certain types of graphs, these encodings may either not provide significant benefits or may be computationally infeasible. In such cases, alternative forms of Higher-Order Graph Encodings can be explored based on the new scenario, as ExGRG provides a flexible framework for incorporating this prior information into the SSL process. Additionally, even when such prior information is not available, other methods for constructing the relation graph—such as using Neighborhood Similarity (Section 3.3.1) and Deep Clustering (Section 3.3.3)—remain applicable and beneficial.

---

> ### Author Response · Authors · 2024-09-14
> **W2**
>
> **W2. It's unclear whether the prior information indeed resolves the problem;**
>
> ExGRG introduces a novel approach to SSL for graph-structured data by moving away from sole dependence on graph augmentations, which have been shown to be both semantic-altering and counterintuitive. Instead, we propose the explicit generation of a compositional relation graph as an alternative. This approach integrates both prior domain knowledge and online extracted information into the SSL invariance objective.
>
> Section 3.2 outlines the theoretical foundation for this relation graph generation, while Section 3.3 provides a detailed explanation of the various strategies employed for generating these graphs. In Section 4.2, we demonstrate the experimental superiority of our method compared to existing graph SSL baseline approaches.

---

> ### Author Response · Authors · 2024-09-14
> **W3**
>
> **W3. There are no experiments evaluating the hyperparameter sensitivity in terms of the objective Eq 10;**
>
> Due to space limitations, we only presented the effect of varying $\alpha_1$ in Eq. 10, as shown in Fig. 13 and discussed in Section B.7.3, "Hyperparameter Analysis," in the appendix. However, we have conducted extensive experiments to analyze the sensitivity with respect to the other loss coefficients as well. Including these results in the paper is an excellent suggestion, and we will do so if space permits. Our findings indicate similar stability across all hyperparameters, showing minimal impact on performance over a range of values.
>
> Additionally, as shown in Fig. 11–13, we illustrate the stability of downstream performance across a broad spectrum of hyperparameter values. This suggests that the overall performance is not overly sensitive to precise hyperparameter choices, allowing for a wide acceptable range of values.

---

> ### Author Response · Authors · 2024-09-14
> **W4**
>
> **W4. The accuracies in Cora and CiteSeer are significantly higher than those of the supervised approaches. It's suggested to check the code and reproducibility;**
>
> We have thoroughly re-evaluated our implementation after observing this performance gap. To enhance reproducibility and to identify any potential platform defects, we are considering open-sourcing ExGRG.
>
> As outlined in Section 4.2, paragraph two, point (ii), we propose the following explanation for the observed performance improvement. When comparing MLP and GCN, the performance gap is more evident in the Cora and CiteSeer datasets than in CoCS and CoPhy. This highlights the critical role of graph structural properties leveraged by the message-passing mechanism. Notably, this trend corresponds to the datasets where ExGRG exhibits the most significant improvements over the state-of-the-art, underscoring the model's effectiveness in integrating adjacency information and positional structural encodings (PSEs) through the use of relation graphs.

---

> ### Author Response · Authors · 2024-09-14
> **Ch1**
>
> **Ch1. The incorporated prior information may not be applicable to all graph SSL scenarios, for example, heterophilic graphs or graph-level classification. In fact, the prior information can be directly incorporated into supervised learning to quickly learn the desired representations instead of taking such a complicated way;**
>
> For alternative methods of constructing relation graphs in the absence of prior information, please refer to W1. Additionally, we investigated the performance of our proposed model without prior information in the ablation studies presented in Tables 2, 4, and 5 under the condition "No $ G^{PSE}$ or $G^{A'}$". Even under this scenario, our model demonstrates superior performance compared to both supervised methods and the SotA SSL approaches. Further details on this analysis can be found in Section 4.3.

---

> ### Author Response · Authors · 2024-09-14
> **Ch2.1**
>
> **Ch2.1. It's unclear whether the prior information indeed resolves the problem.**
>
> Please refer to W2.

---

> ### Author Response · Authors · 2024-09-14
> **Ch2.2**
>
> **Ch2.2. For the analysis based on LE, why original augmentation-based approaches can be considered as an LE optimization, if they are not using VICReg?**
>
> As outlined in Section 3.2.1, optimizing the variance-covariance-constrained VICReg objective is equivalent to solving the Laplacian Eigenmap (LE) problem. We choose to avoid representation collapse by using variance-covariance terms rather than relying on negative samples, as done in methods like SimCLR. In this section, we elaborate on how constructing an explicit relation graph, as opposed to relying on augmentations, parallels the operation of LE and how ExGRG helps mitigate Laplacian rank deficiency.
>
> Other approaches, such as SimCLR and BYOL, can be viewed as some other instances of spectral embedding methods like Multidimensional Scaling or Kernel Canonical Correlation Analysis, rather than LE [1]. Although ExGRG is motivated by VICReg, the process of constructing the relation graph could potentially be adapted for SimCLR or BYOL. However, the theoretical implications of such adaptations would need to be reconsidered and are beyond the scope of this work. This direction could be explored in future research.
>
> [1] Balestriero, Randall, and Yann LeCun. "Contrastive and non-contrastive self-supervised learning recover global and local spectral embedding methods." Advances in Neural Information Processing Systems 35 (2022): 26671-26685.

---

> ### Author Response · Authors · 2024-09-14
> **Ch2.3**
>
> **Ch2.3. Moreover, how does the EM-based analysis imply the advantages of ExGRG over previous augmentation-based approaches?**
>
> Section 3.2.2 explores how the ExGRG procedure can be interpreted from an Expectation-Maximization (EM) optimization perspective. In this discussion, we do not claim any inherent superiority over augmentation-based methods. Rather, we emphasize the difference in explicitly formulating and solving the self-supervised learning (SSL) process through EM, as opposed to the more common implicit augmentation-based approaches. This discussion, along with the perspective of Laplacian Eigenmaps (LE), provides the theoretical foundation for our proposal, ensuring that it is not merely a heuristic or an ad hoc algorithm.

---

> ### Author Response · Authors · 2024-09-14
> **Ch3**
>
> **Ch3. The accuracies in Cora and CiteSeer are significantly higher than those of the supervised approaches. It's suggested that the code and reproducibility be checked by the reviewers.**
>
> Please refer to W4.

---

> ### Author Response · Authors · 2024-09-14
> **Ch4**
>
> **Ch4. There are no experiments evaluating the hyperparameter sensitivity in terms of the objective Eq 10;**
>
> Please refer to W3.

---

> ### Comment · Reviewer_cKEx · 2024-09-15
>
> Thanks to the authors for the explanation. Some of my concerns are addressed. However, many of my questions are not directly and clearly explained with evidence.
>
> For Q1, there's no evidence showing how to mitigate the missing prior information in the node classification scenarios, and in heterophilic graphs, or graph-level classification. Moreover, no baseline is provided for why incorporating the prior information in SSL is better than directly adopting it.
>
> For Q2, it's still unclear why the proposed method is better than the augmentation-based one. The authors claim that they do not intend to demonstrate superiority, which does not align with the motivation of this work.
>
> For the remaining questions, it would be appreciated if there is specific evidence provided for checking, such as experimental results and codes.

---

> > ### Author Response · Authors · 2024-09-15
> > **Q1 & Q2**
> >
> > We sincerely appreciate your timely and thorough review of our work. We are committed to addressing the concerns raised and providing further clarification. Your valuable feedback and the time dedicated to reviewing our work are greatly appreciated.
> >
> > **For Q1, there's no evidence showing how to mitigate the missing prior information in the node classification scenarios and in heterophilic graphs or graph-level classification.**
> >
> > As noted in our response to W1 and W2, the term "prior information" refers to the second strategy discussed in Section 3.3.2, which covers Higher-Order Graph Encodings, specifically Positional and Structural Encodings (PSEs). We demonstrate that integrating these PSEs, using our proposed approach for constructing relation graphs, is highly effective across the nine node classification datasets commonly utilized in graph SSL research.
> >
> > As highlighted, our ablation studies confirm the superior performance of our model even when this prior information strategy is omitted, while our other two key strategies (outlined in Sections 3.3.1 and 3.3.3) remain intact. This is analogous to the scenarios you mentioned, where PSEs might not yield significant advantages or could become computationally prohibitive.
> >
> > **Moreover, no baseline is provided for why incorporating the prior information in SSL is better than directly adopting it.**
> >
> > In Section 3.3.2, we propose a novel approach for utilizing PSEs to construct relation graphs that guide the invariance term during training. As you suggested, these PSEs could alternatively be integrated directly into graph transformers or used as additional input features for GCNs. Current research continues to explore effective ways to incorporate PSEs directly into graph representation learning, particularly in architectures like graph transformers, with the goal of surpassing GCN performance [1, 2]. However, this direction is beyond the scope of our work.
> >
> > [1] Dwivedi, Vijay Prakash, Anh Tuan Luu, Thomas Laurent, Yoshua Bengio, and Xavier Bresson. "Graph neural networks with learnable structural and positional representations." arXiv preprint arXiv:2110.07875 (2021).
> >
> > [2] Cantürk, Semih, Renming Liu, Olivier Lapointe-Gagné, Vincent Létourneau, Guy Wolf, Dominique Beaini, and Ladislav Rampášek. "Graph positional and structural encoder." arXiv preprint arXiv:2307.07107 (2023).
> >
> >
> > **For Q2, it's still unclear why the proposed method is better than the augmentation-based one. The authors claim that they do not intend to demonstrate superiority, which does not align with the motivation of this work.**
> >
> > We believe there may be a misunderstanding regarding our response to Ch2.3. As clarified, the EM-based analysis (Section 3.2.2) does not suggest any inherent superiority of our model over augmentation-based methods. However, we maintain our claim of superiority over augmentation-based SSL approaches, as graph augmentations have been shown to be both semantically disruptive and counterintuitive.
> >
> > Additionally, Section 3.2.1 reinforces our argument by providing an LE-based perspective. It explains how the use of explicit relation graphs, as opposed to implicit augmentation-based one, helps mitigate Laplacian rank deficiency and establishes connections between otherwise disconnected islands. ExGRG moves away from relying solely on graph augmentations, instead proposing the explicit generation of a compositional relation graph as an alternative.

---

### Review · Reviewer_tn9u · 2024-09-02

**Summary Of Contributions:**

This work proposes a non-contrastive graph SSL approach to address the issues of semantic-altering and counter-intuitive nature of graph augmentations.

**Audience:**

Yes

**Broader Impact Concerns:**

I did not identify any concerns about the ethical implications of the work.

**Claims And Evidence:**

Yes

**Requested Changes:**

Please refer to the weaknesses listed above.

**Strengths And Weaknesses:**

# Strengths

- The idea of generating relation graphs is interesting.
- The techniques seem to be sound.
- The experiments are extensive, and the results seem to be promising.

# Weaknesses

- The motivation is weak since there exist some graph SSL methods [1] that do not require data augmentation.
- It would be better to provide the URL of source codes and datasets to facilitate better reproducibility of this work.
- In Section 2.2, the investigation of the graph representation learning methods is not comprehensive, since some recent relate works [2,3] are missing.

*Refs*:

[1] "Self-supervised heterogeneous graph pre-training based on structural clustering." Advances in Neural Information Processing Systems 35 (2022): 16962-16974.

[2] "Interpretable and efficient heterogeneous graph convolutional network." IEEE Transactions on Knowledge and Data Engineering 35.2 (2021): 1637-1650.

[3] "Graph substructure assembling network with soft sequence and context attention." IEEE Transactions on Knowledge and Data Engineering 35.5 (2022): 4894-4907.

---

> ### Author Response · Authors · 2024-09-14
> **Appreciation for Your Valuable Feedback**
>
> Thank you for your time and effort in providing valuable feedback on our work. We have carefully considered your comments and addressed each of the points raised below.

---

> ### Author Response · Authors · 2024-09-14
> **W1**
>
> **W1. The motivation is weak since there exist some graph SSL methods [1] that do not require data augmentation.**
>
> SHGP [1] is tailored explicitly for Heterogeneous Information Networks (HINs) and has been evaluated using benchmark datasets specific to HINs. In contrast, ExGRG is designed for more general graph self-supervised learning (SSL) tasks. We believe the introduction of methods like AFGRL [4], SPGCL [5], and ExGRG highlights the significant motivation within the research community to develop approaches that do not depend exclusively on graph augmentations, which can be semantic-altering or counterintuitive.
>
> [1] Yang, Yaming, Ziyu Guan, Zhe Wang, Wei Zhao, Cai Xu, Weigang Lu, and Jianbin Huang. "Self-supervised heterogeneous graph pre-training based on structural clustering." Advances in Neural Information Processing Systems 35 (2022): 16962-16974.
>
> [4] Lee, Namkyeong, Junseok Lee, and Chanyoung Park. "Augmentation-free self-supervised learning on graphs." In Proceedings of the AAAI conference on artificial intelligence, vol. 36, no. 7, pp. 7372-7380. 2022.
>
> [5] Wang, Haonan, Jieyu Zhang, Qi Zhu, Wei Huang, Kenji Kawaguchi, and Xiaokui Xiao. "Single-pass contrastive learning can work for both homophilic and heterophilic graph." arXiv preprint arXiv:2211.10890 (2022).

---

> ### Author Response · Authors · 2024-09-14
> **W2**
>
> **W2. It would be better to provide the URL of source codes and datasets to facilitate better reproducibility of this work.**
>
> Thank you for your valuable feedback. This is an excellent suggestion, and we will take it into consideration for the final version of the paper.

---

> ### Author Response · Authors · 2024-09-14
> **W3**
>
> **W3. In Section 2.2, the investigation of the graph representation learning methods is not comprehensive, since some recent relate works [2,3] are missing.**
>
> Thank you for your feedback. While the two referenced works [2, 3] address tasks different from those of ExGRG, they present interesting insights and will be included in the final version of the paper.
>
> [2] Yang, Yaming, Ziyu Guan, Jianxin Li, Wei Zhao, Jiangtao Cui, and Quan Wang. "Interpretable and efficient heterogeneous graph convolutional network." IEEE Transactions on Knowledge and Data Engineering 35, no. 2 (2021): 1637-1650.
>
> [3] Yang, Yaming, Ziyu Guan, Wei Zhao, Weigang Lu, and Bo Zong. "Graph substructure assembling network with soft sequence and context attention." IEEE Transactions on Knowledge and Data Engineering 35, no. 5 (2022): 4894-4907.

---

> ### Author Response · Authors · 2024-09-27
> **Revision**
>
> **W3:**
> We uploaded a revised manuscript and addressed the concern regarding missing related works by incorporating two recently published studies into Section 2.2.

---

### Decision · Action_Editor_Uhfh · 2024-10-16

**Recommendation:** Reject

**Comment:**

Besides the issues mentioned above around claims and evidence, reviewers further pointed out the following:
- The proposed method is quite complex and  has a lot of hyperparameters which may be difficult to tune.
- All reviewers found clarity issues remaining after the latest update.

**Audience:**

Tthis paper would be of interest to members of TMLR's audience working on self-supervised learning or graph representation learning.

**Claims And Evidence:**

The reviewers gave mixed recommendations for the paper. The AE decided to side with the majority of reviewers that recommended (weak) rejection, and raised the following issues regarding claims and evidence:
- More experimental evidence seems to be required for some cases (e.g. more general node classification benchmarks and heterophilic graphs)
- The reviewers further had concerns around reproducibility, since results on datasets like Cora are significantly higher than related works.
- More convincing evidence is required to showcase the advantages of ExGRG over previous augmentation-based approaches, theoretical or otherwise.

**Resubmission Of Major Revision:**

The authors may consider submitting a major revision at a later time.